# Late-life dietary folate restriction reduces biosynthesis without compromising healthspan in mice

Heidi M Blank[1], Staci E Hammer[1], Laurel Boatright[1,2], Courtney Roberts[1], Katarina E Heyden[3], Aravindh Nagarajan[2,4], Mitsuhiro Tsuchiya[5], Marcel Brun[6], Charles D Johnson[6], Patrick J Stover[1,7,13], Raquel Sitcheran[8], Brian K Kennedy[9,10], L Garry Adams[11], Matt Kaeberlein[5,12], Martha S Field[3], David W Threadgill[1,4,13,14], Helene L Andrews-Polymenis[2,4], Michael Polymenis[1,4,7]

**Folate is a vitamin required for cell growth and is present in fortified foods in the form of folic acid to prevent congenital abnormalities. The impact of low-folate status on life-long health is poorly understood. We found that limiting folate levels with the folate antagonist methotrexate increased the lifespan of yeast and worms. We then restricted folate intake in aged mice and measured various health metrics, metabolites, and gene expression signatures. Limiting folate intake decreased anabolic biosynthetic processes in mice and enhanced metabolic plasticity. Despite reduced serum folate levels in mice with limited folic acid intake, these animals maintained their weight and adiposity late in life, and we did not observe adverse health outcomes. These results argue that the effectiveness of folate dietary interventions may vary depending on an individual's age and sex. A higher folate intake is advantageous during the early stages of life to support cell divisions needed for proper development. However, a lower folate intake later in life may result in healthier aging.**

## Introduction

Folate is a metabolic cofactor that transfers single-carbon functional groups (one-carbon units; 1C; see Fig 1A) within a metabolic network known as folate-mediated one-carbon metabolism (FOCM). These reactions are involved in the metabolism of amino acids (primarily Ser, Gly, Met, and His), the synthesis of purines, thymidylate, and phospholipids and numerous methylation reactions (West et al, 1996; Fox & Stover, 2008; Locasale, 2013; Labuschagne et al, 2014; Ducker & Rabinowitz, 2017; Rosenzweig et al, 2018; Zheng & Cantley, 2019; Reina-Campos et al, 2020). Hence, folate is vital for cell division, especially during rapid proliferation states, including fetal development. Conversely, antifolate drugs have been used for decades to inhibit excessive cell proliferation in cancer, rheumatoid arthritis, and psoriasis. To help prevent congenital disabilities, the USA and other countries have implemented large-scale folate fortification of staple foods. The benefits of dietary folate early in life are well-studied and documented. In contrast, its role later in life in modulating the healthy, disease-free period of an individual's life (a.k.a. healthspan) is poorly understood.

Concerns have been raised that mandatory folic acid fortification could negatively affect older individuals. Some observations suggested a possible link between folic acid fortification and increased colorectal cancer rates in the 1990s (Mason et al, 2007). However, later analyses "did not identify specific risks from existing mandatory folic acid fortification" in the general population (Field & Stover, 2018). This conclusion neither refutes nor contradicts the idea that a moderate decrease in folic acid intake among older adults may improve healthspan. Merely because high folic acid intake does not harm the health of older adults does not negate the possibility that a lower folic acid intake might enhance health.

Previous work from several labs, including ours (Maitra et al, 2020), suggested that loss-of-function genetic interventions in 1C enzymes promote longevity in invertebrate model systems. Lower methyltetrahydrofolate (methyl-THF) level is a common signature

[1]Department of Biochemistry and Biophysics, Texas A&M University, College Station, TX, USA   [2]Department of Microbial Pathogenesis and Immunology, School of Medicine, Texas A&M University Health Science Center, Bryan, TX, USA   [3]Division of Nutritional Sciences, Cornell University, Ithaca, NY, USA   [4]Interdisciplinary Program in Genetics, Texas A&M University, College Station, TX, USA   [5]Department of Laboratory Medicine and Pathology, University of Washington, Seattle, WA, USA   [6]Texas A&M Agrilife Research, Genomics and Bioinformatics Service, College Station, TX, USA   [7]Institute for Advancing Health Through Agriculture, Texas A&M University, College Station, TX, USA   [8]Department of Cell Biology and Genetics, School of Medicine, Texas A&M University Health Science Center, Bryan, TX, USA   [9]Departments of Biochemistry and Physiology, Yong Loo Lin School of Medicine, National University of Singapore, Singapore, Singapore   [10]Healthy Longevity Translational Research Programme, Yong Loo Lin School of Medicine, National University of Singapore, Singapore, Singapore   [11]Department of Veterinary Pathobiology, College of Veterinary Medicine, Texas A&M, College Station, TX, USA   [12]Optispan, Inc., Seattle, WA, USA   [13]Department of Nutrition, Texas A&M University, College Station, TX, USA   [14]Texas A&M Institute for Genome Sciences and Society, Texas A&M University, College Station, TX, USA

Correspondence: michael.polymenis@ag.tamu.edu

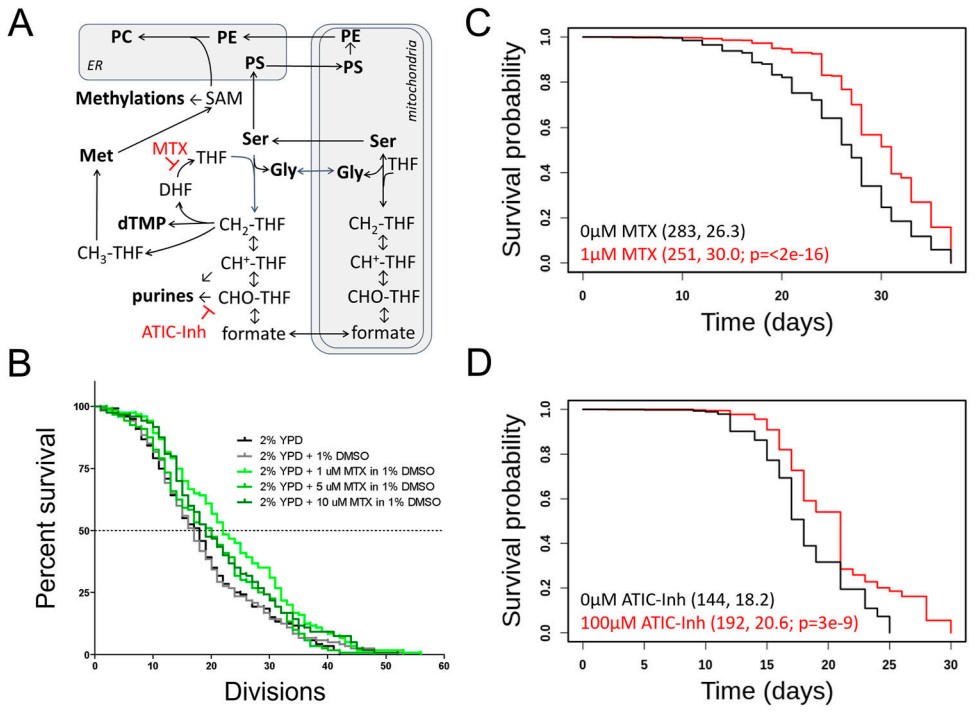

**Figure 1. Inhibitors of 1C metabolism extend the lifespan of yeast and worms.**
**(A)** Schematic of 1C enzymatic reactions. The reactions inhibited by methotrexate (MTX) or ATIC Dimerization Inhibitor are indicated.
**(B)** Survival curves on rich undefined media for *S. cerevisiae* MATα (strain BY4742) cells (shown in black), compared with experiment-matched cells mock-treated with DMSO (shown in gray) or three different doses of MTX (as indicated, in different shades of green). The number of mother cell divisions (replicative lifespan) is on the x-axis.
**(C)** Survival curves for *C. elegans* (strain N2) exposed to the indicated doses of MTX. Survival probability is on the y-axis, and time (in days) is on the x-axis. Mean lifespans and the number of animals assayed in each case are shown in parentheses. The indicated *P*-value was based on the log-rank test.
**(C, D)** Survival curves, as in (C), for animals exposed to the indicated doses of ATIC Dimerization Inhibitor.
Source data are available for this figure.

of pro-longevity pathways (Annibal et al, 2021), and methionine restriction extends longevity in several organisms (Orentreich et al, 1993; Miller et al, 2005; Sun et al, 2009; Cabreiro et al, 2013; Lee et al, 2014, 2016; Johnson & Johnson, 2014; Kozieł et al, 2014; Ruckenstuhl et al, 2014; Barcena et al, 2018; Zou et al, 2020). Mutations in the large (60S) subunit ribosomal proteins constitute a significant class of pro-longevity mutations in yeast and other species (Kaeberlein et al, 2005; Hansen et al, 2007; Steffen et al, 2008, 2012; Kaeberlein & Kennedy, 2011; McCormick et al, 2015). Using ribosome profiling, we identified translational control patterns responsible for increased longevity. Long-lived mutants had significantly reduced translational efficiency of transcripts encoding enzymes of the broader network of 1C metabolism (Maitra et al, 2020). Steady-state metabolite profiling of these mutants was consistent with changes in 1C metabolism (Maitra et al, 2020).

The effects of folate intake in aged adults have yet to be studied in mouse models, where causal mechanisms can be examined more precisely. In this report, we examined healthspan as a function of dietary folate intake in the long-lived inbred mouse strain C57BL/6J. We found that restricting folate intake in aged mice does not adversely affect their health and may even improve their metabolic plasticity, as they switch between glycolysis and oxidative phosphorylation states during their regular diurnal cycles. Metabolite changes in the folate-restricted animals suggest reduced anabolism. Similarly, the abundance of transcripts encoding gene products involved in protein synthesis was reduced in animals limited for folate late in life. These results are consistent with a view of 1C metabolism as a tunable platform that allocates cellular resources for biosynthesis. They also suggest that the outcomes of folate dietary interventions might depend on an individual's age. Adequate folate status is required during development for neural

tube closure. Late in life, a lower folate intake might promote healthy aging.

# Results

## Inhibitors of 1C metabolic enzymes extend the lifespan of yeast and worms

We reasoned that if genetic interventions of 1C metabolism extend yeast's lifespan, chemical ones may also. Methotrexate leads to tetrahydrofolate (THF) limitation (Rajagopalan et al, 2002). Methotrexate has been used extensively since its introduction in the 1950s but has not been evaluated for longevity effects. We found that methotrexate (at 0.5–10 *μ*M) increased yeast replicative lifespan (Fig 1B, *P* < 0.05 based on the log-rank test). The dose with the maximal (~15%) lifespan extension was 1 *μ*M (Fig 1B). At such low doses, methotrexate did not affect cell proliferation significantly (not shown). We also looked at cell size as a proxy of cell cycle effects. Folate deficiencies and drugs that interfere with DNA replication are often associated with increased cell size, leading to megaloblastosis (Scott & Weir, 1980; Das et al, 2005). Because methotrexate inhibits dihydrofolate reductase, it limits the levels of THF for thymidylate synthesis and DNA replication (Rajagopalan et al, 2002). At concentrations up to 10 *μ*M, methotrexate increased cell size slightly (from 47 to 52 fL; Fig S1). At higher concentrations, methotrexate did not extend replicative longevity (Fig 1B) and increased cell size more significantly (from 47 to 56–64 fL, for 50–100 *μ*M, respectively; Fig S1), indicating a stronger cell cycle block. These data are consistent with the notion that moderate, but not severe, cell cycle delays

are associated with longer replicative lifespan in yeast (He et al, 2014).

Given the conservation of 1C pathways, we asked if methotrexate could increase the lifespan of wild-type *Caenorhabditis elegans* animals. It has been reported that treating worms with high levels of methotrexate (220 µM) at the adult stage did not extend their lifespan (Virk et al, 2016). Because adult worms are post-mitotic, we exposed *C. elegans* to the drug continuously, from the embryo stage until death (Fig 1C). Worms exposed to low doses of methotrexate (1–3 µM) had a longer lifespan (~15% lifespan extension, $P < 2 \times 10^{-16}$ based on the log-rank test, Fig 1C). At higher doses (10–100 µM), methotrexate did not extend lifespan (not shown), in agreement with (Virk et al, 2016), who treated adult animals with 220 µM methotrexate. We also note that the bacteria used to feed the worms in our experiments were killed by ultraviolet radiation to exclude any impacts from bacterial folate metabolism, which is known to affect worm lifespan (Virk et al, 2012, 2016). We conclude that the longevity extension by methotrexate is conserved between yeast and invertebrates.

To further test the generality of the idea that 1C interventions promote longevity, we examined another commercially available inhibitor of a 1C enzyme. A dipeptide that blocks the requisite dimerization of ATIC (AICAR Transformylase/Inosine Mono-phosphate Cyclohydrolase; see Fig 1A) has been previously shown to activate the AMPK signaling pathway and ameliorate the metabolic syndrome in mice (Asby et al, 2015). In experiment-matched assays, the ATIC inhibitor also increased the lifespan of worms at 100 µM (Fig 1D). Without inferring any equivalence in outcomes across species, we note that this dose is similar to the one used in mice with metabolic syndrome (Asby et al, 2015).

To our knowledge, the above is the first evidence arguing that pharmacologic interventions of 1C metabolism increase the lifespan of invertebrate model organisms, raising the exciting possibility that 1C interventions may improve longevity in mammals. Acute toxicity of methotrexate was measured in numerous studies, starting in 1950 with measurements in mice, rats, and dogs (Ferguson et al, 1950; Philips et al, 1950). Only later, in the 1970s, rodents were subjected to low-dose, long-term treatment with methotrexate to measure its toxicity (Rustia & Shubik, 1973; Freeman-Narrod & Narrod, 1977). Methotrexate given to 7-wk-old Syrian hamsters for several months was tolerated well at 5, 10, and 20 ppm (Rustia & Shubik, 1973). Methotrexate was also tolerated quite well in mice when administered at 3, 5, 8, or 10 ppm in the diet (Rustia & Shubik, 1973) or by IP injection at 0.25–2 mg/kg (Freeman-Narrod & Narrod, 1977). We reasoned that those long-term studies might offer helpful information about the role of folate limitation in longevity in mice. We generated survival curves from the available historical data (0–10 ppm MTX) given in alternating weeks in the diet in male and female Swiss mice from 7 to 120 wk of age. In five of the eight conditions tested, the mean lifespan was longer than in the control group. In one case, it was significantly so ($P = 0.04$, based on the log-rank test, Fig S2). The number of females used was too low to detect significant differences in lifespan and no healthspan parameters were evaluated in those studies (Rustia & Shubik, 1973). These limitations notwithstanding, the results are significant because the drug was administered from a young age (7 wk) when side effects are expected to be more pronounced. The LD50 for

methotrexate given to 5-wk-old mice is 59 mg/kg, whereas that for 16-wk-old mice is 284 mg/kg (Freeman-Narrod & Narrod, 1977). Based on these observations, and our results in yeast and worms (Fig 1) we decided to measure if folate limitation later in life improves healthspan in mice.

## Mice placed under dietary folate restriction late in life maintain or increase their weight and do not develop anemia

Because relatively little is known about healthspan as a function of dietary folate intake, we designed a study in the long-lived inbred strain C57BL/6J (Yuan et al, 2009) (Fig 2A, and see the Materials and Methods section). Longevity was not the measured outcome; the mice were euthanized at 120 wk. However, with a sample size of 20 mice per group, there was sufficient power to detect significant ($P < 0.05$ and 80% power) differences (1SD) in healthspan parameters (Bellantuono et al, 2020). Starting at 52 wk of age, half the mice were maintained on the standard diet (AIN-93M; [Reeves et al, 1993]), and the other half were placed on a folate/choline-deficient (F/C–) diet. Before the diet changes at 52 wk of age, the mice were randomly assigned to the different test groups based on their lean mass. The standard diet contains 2 mg/kg of folic acid and 2.5 g/kg choline bitartrate (F/C+), whereas the F/C– diet contains 0 mg/kg of folic acid and 0 mg/kg choline bitartrate. As expected, serum folate levels were greatly reduced in the F/C– groups ($P = 0.00216$ for the females and $P = 0.00012$ for the males, based on the Wilcoxon rank sum test; Fig 2B).

In people, folate supplementation during pregnancy is positively associated with birth weight (Scholl & Johnson, 2000), but in older adults there is no association between folate status and weight (Soysal et al, 2019). We found that the weight of mice from either sex was not reduced from 52 wk of age when placed on the F/C– diet until the end of the study at 120 wk of age (Fig 1C). Instead, male mice on the F/C– diet gained weight (Fig 1C, compare the two left panels). To evaluate the statistical significance of this observation, we applied a mixed effects regression model to analyze the repeated longitudinal weight measurements using the lme4 and lmer R language packages. The model was valid because from a scatter plot of the standardized residuals versus the fitted values, the residuals were symmetrically distributed around zero, with approximately constant variance (Fig S3A). Furthermore, the normality of the residuals was checked with a quantile-quantile plot (Fig S3B). Based on the mixed effects regression model, there was a negative effect between diet (F/C+) and weight (slope = –1.96, $P = 0.0313$). As expected, there was a small but significant positive relation between weight and time (slope = 0.075, $P < 2 \times 10^{-16}$) and a strong one between weight and male sex (slope = 7.72, $P = 6.48 \times 10^{-13}$).

The mice on the F/C– diet were not anemic (Fig S4). They had the same blood cell counts as the mice on the F/C+ diet (Fig S4A). There were also no cell size changes or evidence of megaloblastosis (Fig S4B), which one might expect if DNA replication in erythrocytes was significantly inhibited (Das et al, 2005). Lastly, mice on the F/C– diet did not have reduced survival compared with animals of the same sex that were kept on the F/C+ diet (Fig S5). It is unclear why the female animals had reduced survival compared with males. Still, the increased mortality of C57BL/6J females is in line with data from the Aged Rodent Colonies maintained by the National Institute on

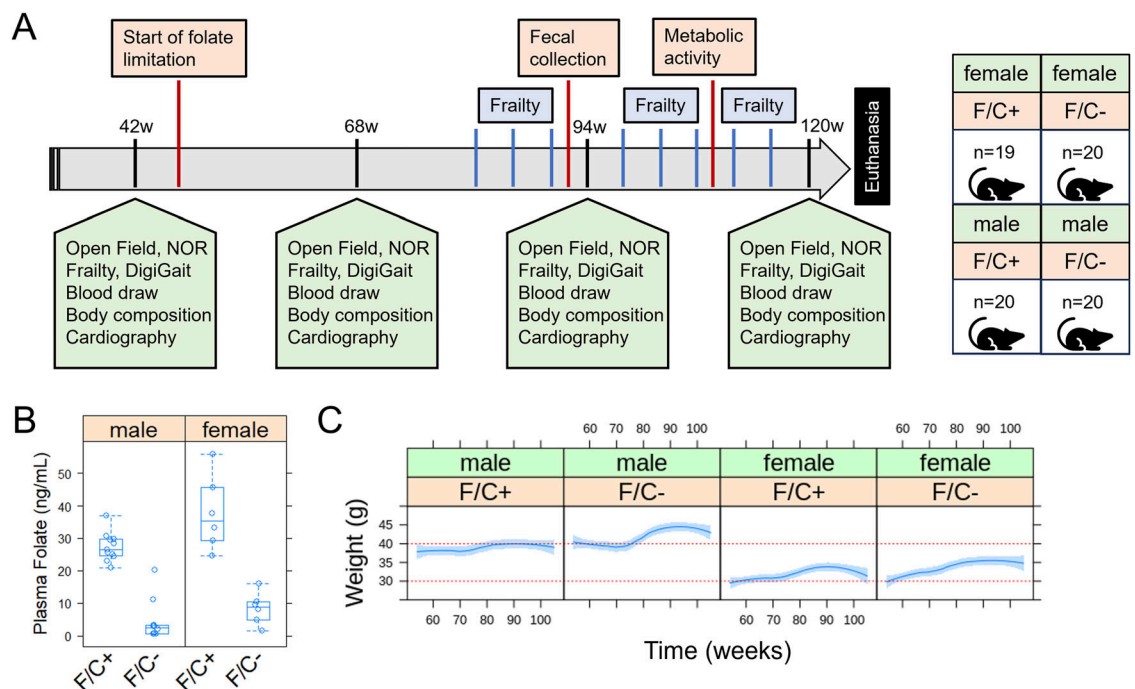

**Figure 2. Evaluating healthspan in mice under dietary folate restriction late in life.**
**(A)** Schematic of the study design. See the Materials and Methods section for a detailed description of each assay. **(B)** Serum folate levels were measured at 120w, using an established microbiological assay (Cooper & Jonas, 1973) and shown on the y-axis. The different diets are on the x-axis, as indicated for the folate/choline-replete (F/C+) or -limited (F/C−) groups. The boxplot graphs were generated with R language functions. Each box is drawn from the first to the third quartile, with a horizontal line denoting the median. The whiskers show the interquartile range, and they were drawn at 1.5 x interquartile range. The replicates were all biological ones from different animals. **(C)** The weight of the animals (y-axis) was measured every month (x-axis). The red horizontal lines were drawn to help visualize the weight changes over time in female and male mice. Loess curves and the std errors at a 0.95 level are shown. A mixed effects regression model was applied with the *lme4 and lmer* R language packages to evaluate the effects of the fixed variables (diet, time, sex) on the observed weight, taking into account the repeated longitudinal measurements on each mouse (see the Materials and Methods section, and description in the text). A negative association with the F/C+ diet was significant ($P$ = 0.0313). Source data are available for this figure.

Aging. Females experience 50–66% mortality between 20 and 30 mo of age versus 30–36% mortality for male mice (Turturro et al, 1999). These data show that dietary folate limitation late in life does not lead to anemia, reduced viability, or reduced body weight. In contrast, at least in the case of males, folate-limited animals have a higher body weight.

**No adverse healthspan metrics in mice placed on dietary folate restriction late in life**

At the indicated times shown in Fig 2A, we evaluate various metrics associated with healthspan. Mice on the F/C− diet had similar Frailty Index scores ($P$ = 0.434, based on a mixed effects regression model) with their counterparts on the F/C+ diet (Fig 3A). The Frailty Index is a clinically validated metric comprising several visible clinical signs of physical deterioration, as previously described for aging C57BL/6J mice (Whitehead et al, 2014; Kane et al, 2016). Note that at the "<52w" timepoint, the diet had not been switched yet, and all the mice were on the F/C+ diet. However, to accurately track individual animals and visualize the data, the mice that were placed in the different diet groups at 52 wk of age are also depicted in the same groups at the "<52w" time point.

In addition to the regular body weight measurements described in Fig 2, we also evaluated body composition based on

measurements with EchoMRI (Fig 3B). There was again a significant negative effect from the F/C+ diet on total body mass (slope = −2.1674, $P$ = 0.04580, based on a mixed effects regression model). There was also a negative trend between the F/C+ diet and fat mass (slope = −1.6989, $P$ = 0.0635), and a weaker negative association with lean mass (slope = −0.5034, $P$ = 0.1474), although in the latter cases the effects did not reach the $P < 0.05$ threshold.

Regarding the other healthspan-related metrics we evaluated at 68, 94, and 120 wk of age (see Fig 2A and see the Materials and Methods section), there were no significant diet effects based on mixed effects models. The metrics we measured included gait analysis during voluntary walking, which evaluates a wide range of ambulatory problems, using the accurate and high throughput DigiGait system (Fig S6), open field (Fig S7A), and novel object recognition parameters (Fig S7B), which evaluate cognitive behavior and memory. In the case of novel object recognition, in addition to the discrimination index (DI) values shown in Fig S7B, we also evaluated differences in a binary, pass-fail format. We chose a DI > 0.06 as the cutoff for a passing novel object test, as this also correlated visually with what was seen on a heat map of exploration around each object. Based on $\chi^2$ tests, there were again no significant diet effects. Lastly, cardiac function was evaluated with echocardiography using the Vevo 3100 Ultrasound device, looking specifically for aging-related changes in cardiac physiology

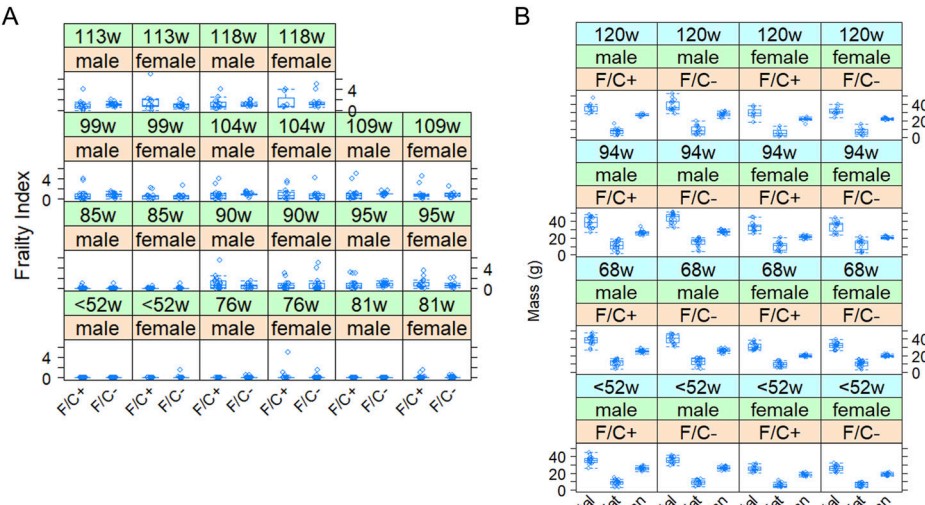

Figure 3. **No adverse healthspan metrics in mice placed on dietary folate restriction late in life.** **(A)** Frailty Index scores are shown on the y-axis. Measurements were taken at the indicated times from female and male animals of each diet test group, as described in the Materials and Methods section. **(B)** The total, fat, and lean mass of each mouse in the study was measured by MRI (see the Materials and Methods section), and shown on the y-axis. The boxplots were drawn as in Fig 2. Source data are available for this figure.

(Lindsey et al, 2018). Cardiac output, systole and diastole diameter, ejection fraction, and fractional shortening were all unaffected by diet (Fig S8).

## Metabolic activity of aged mice on the folate-restricted diet

Not only were no healthspan metrics adversely affected by limiting dietary folate intake late in life (Fig 3), but we also noticed some positive outcomes. At ~85 wk of age, the male animals on the folate-limited diet were visibly less gray than their counterparts on the folate-replete diet (Fig 4A). Although graying is not necessarily associated with declining health, it is usually an age-dependent trait. Notably, recent work in human cells showed that inhibiting the target of rapamycin (TOR) inhibits graying (Suzuki et al, 2023), consistent with the idea that delayed hair graying may be an outcome of interventions that improve healthspan.

Because folate-based 1C chemical reactions are a metabolic hub (see Fig 1A), we placed individual animals in TSE Phenomaster metabolic cages (see the Materials and Methods section). We measured several physiological parameters reporting global metabolic features (Fig 4B). A metric of particular significance is the respiratory exchange ratio (RER), which is the ratio between the metabolic production of carbon dioxide ($CO_2$) and the uptake of oxygen ($O_2$). RER is an indicator of whether carbohydrate or fat is being metabolized. An RER <1 suggests that fat is predominantly used (e.g., during sleep). A value of 1 is indicative of carbohydrate use. RER can exceed 1 during exercise. How fast the RER changes between periods of inactivity versus activity (e.g., during diurnal transitions) reflects metabolic plasticity, which is expected to decline with age. Male animals on the F/C+ diet increased their RER as they became more active during the night (Fig 4B, left bottom panel). The rise in the RER was slower in the female animals on the F/C+ diet (Fig 4B, left top panel), suggesting reduced metabolic plasticity. Their counterparts on the F/C− diet maintained metabolic plasticity, transitioning much faster to carbohydrate-based fuel consumption (Fig 4B, compare the two top panels). Furthermore,

the male mice on the F/C− diet had higher RER values at night than those on the F/C+ diet (Fig 4B, compare the two bottom panels). Based on mixed effects regression models for the period during the transition from daytime to night (time 700–1,100 min in Fig 4B), the effects of time and sex (males) on the RER increase were positive and significant ($P = 1 \times 10^{-6}$ and $P = 0.0101$, respectively), whereas there was a negative association with the F/C+ diet ($P = 0.0806$). These data suggest that a late-life folate-limited diet might have metabolic benefits, albeit for different reasons in the two sexes. Female mice had improved metabolic plasticity, whereas males reached higher RER values.

## Changes in the intestinal microbiome of mice limited for folate late in life

Because the microbiome likely contributes to 1C metabolism, we sampled and sequenced the DNA of the fecal microbiome at 90 wk of age. The analysis yielded standard diversity metrics to assess differences in the fecal microbiome's makeup among the mice groups (Fig 5). The different sex and diet groups had an easily distinguishable gut microbiome, occupying different areas of principal component analysis graphs (Fig 5A), based on Bray-Curtis $\beta$-diversity dissimilarity indices (Knight et al, 2018). The intestinal microbiome of male mice on the F/C− diet was not less diverse ($P = 0.222$, based on the Wilcoxon rank sum test; Fig S9).

We looked at biomarkers from the same metagenomic datasets to place the above differences in the context of 1C metabolism. The LEfSe computational pipeline was used to determine the features most likely to explain the observed differences (Segata et al, 2011). Notably, the LEfSe algorithm incorporates effect sizes, ranking the relevance of the identified biomarkers and enabling their visualization through the computed linear discriminant analysis (LDA) scores (Segata et al, 2011). We found that the microbiome pathway changes were essentially dimorphic for sex. Only enrichment for Coenzyme A biosynthesis was evident in male and female mice on the folate-replete diet (Fig 5B, bottom bar). All other changes were from one sex (to simplify the visualization, we grouped the data

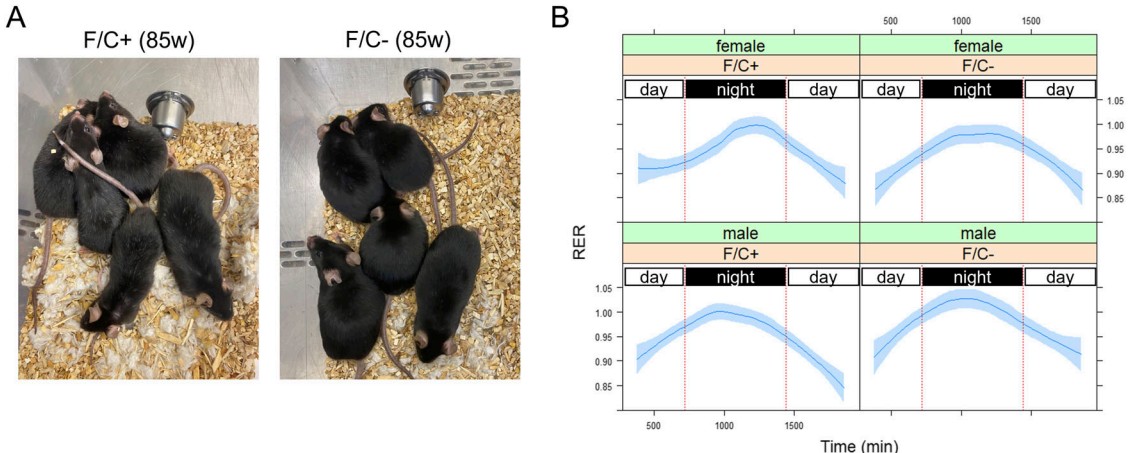

**Figure 4. Improved metabolic activity of mice placed on dietary folate restriction late in life.**
**(A)** Photographs of male mice on the indicated diet were taken at 85 wk of age. Signs of graying were visible on the coat of mice on the F/C+ diet (left) but not on the coat of mice on the F/C− diet (right). **(B)** The respiratory exchange rate values (y-axis) were from six to eight mice in each indicated group at 108 wk of age. The measurements were taken after the animals were acclimated for 2 d in the metabolic cages. The period between the red vertical lines corresponds to night-time when mice are active. Loess curves and the std errors at a 0.95 level are shown.
Source data are available for this figure.

from both sexes). Overall, pathways involved in amino acid (Fig 5B; black arrows) and IMP (Fig 5B; gray arrows) synthesis were enriched in mice on the folate-limited diet. These data suggest that under dietary folate limitation, the microbiome may be a source of metabolites that are known outputs of 1C metabolism (e.g., methionine, and IMP, from which all purine nucleotides are made).

### Metabolite and gene expression changes in the folate-restricted animals consistent with a reduced anabolism

To gauge immune function, we measured the levels of 32 serum cytokines at the time of euthanasia, at 120 wk of age (Fig S10). The differences were again minimal, and those that passed the $P < 0.05$ threshold (indicated with the red asterisks in Fig S11) probably were because of lower variance in the measured values and not to major changes in the levels of those cytokines, comparing animals of the same sex but on different diets. These apparent changes, based on the Wilcoxon rank sum test, were lower IL-15 levels in females on the F/C− diet ($P = 0.0426$); higher IL-17 levels in females on the F/C− diet ($P = 0.0127$); lower VEGF levels in females on the F/C− diet ($P = 0.0237$); and higher LIX levels in males on the F/C− diet ($P = 0.0015$).

Pathologic evaluation of various organs of the euthanized animals at 120 wk of age revealed minimal effects on overall disease burden, other than an apparent increase in kidney abnormalities in male animals on the F/C− diet (Fig S11). Nearly all the mice had some degree/number of perivascular lymphoplasmacytic nodules (Russell bodies), frequently with immunoglobulin aggregates in the ER of plasma cells in the spleen, liver, kidney, and lung. These nodules were higher in number and larger (although only significant in male kidneys) in mice on the F/C− diet, particularly females. This lesion is compatible with but not diagnostic for autoimmune diseases such as systemic lupus erythematosus in aging women.

We note, however, that although the levels of the pro-inflammatory IL-17 were elevated in female mice as well, no leukocytosis was detected.

We next measured signatures associated with genomic instability. Based on targeted bisulfite sequencing at >1,000 loci, there were no changes in the DNA methylation levels from liver samples collected at 120 wk of age (Fig S12A). This analysis yielded an estimate of the DNA methylation age (shown on the y-axis in Fig S12A), which was not different among the sexes and diet groups. The negative values in all groups we examined reflect comparisons with the expected standard values for the strain C57BL/6J and liver tissue we examined, making all our groups appear "younger" than expected based on their DNA methylation profile for unknown reasons. These data suggest that limiting folate late in life did not lead to significant epigenetic changes. We also found that uracil misincorporation was not significantly elevated in liver samples collected at 120 wk of age from mice on the F/C− diet (Fig S12B). Uracil misincorporation into the genome may reflect limited folate availability for DNA synthesis (i.e., if folate is available, then thymidylate synthesis is unperturbed, and uracil is not expected to be incorporated into the DNA). Hence, limiting folate late in life does not lead to genomic instability in mouse liver, though uracil levels are known to vary by tissue in other systems of folate limitation (Chon et al, 2019).

We next sought to identify broad molecular pathway changes associated with folate restriction late in life. First, we measured amino acid levels in the sera of all remaining animals, before they were euthanized. Serum glutamine levels were markedly elevated (~threefold) in male mice on the folate-limited diet (Fig 6A). We did not observe any other significant changes in serum amino acid levels.

Second, we measured the levels of ~600 metabolites from liver samples of all mice that were alive at 120 wk of age by mass spectrometry. In male animals, the metabolite with the lowest relative abundance in folate-limited animals was IMP. Serine had

**Figure 5. Metagenomic profiling of the gut microbiome of mice placed on dietary folate restriction late in life.**
**(A)** Beta-diversity principal component analysis plots were based on Bray-Curtis dissimilarity indices (Knight et al, 2018), of the DNA from the fecal microbiome sampled and sequenced at 90 wk of age, from five mice in each test group. Three principal components (PC1,2,3; shown at the top) accounted for ~70% of the dataset variance. **(B)** Metabolic pathway biomarker changes associated with folate limitation late in life, from metagenomic data of the fecal microbiome. The LEfSe computational pipeline was used to determine the features most likely to explain the observed differences. The computed linear discriminant analysis scores (Log₁₀-transformed) are on the x-axis. Linear discriminant analysis scores incorporate effect sizes, ranking the relevance of the identified biomarkers and enabling their visualization (Segata et al, 2011). Gray arrows indicate pathways involved in amino acid synthesis and black arrows indicate pathways involved in IMP synthesis.
Source data are available for this figure.

the highest relative abundance (Fig 6B). These results are consistent with reduced 1C metabolism because serine is the primary 1C input (Fig 1A). The purine nucleotide IMP is a major output (Fig 1A). Among 589 metabolites detected and assigned, we looked for pathway enrichment with the MetaboAnalyst platform (Chong et al, 2019). There were no significantly overrepresented pathways in either male or female folate-limited animals. In females, pyrimidine metabolism may be under-represented ($P$ = 0.00103, FDR = 0.101). In males, metabolites associated with the pentose phosphate pathway, methionine metabolism, and Warburg effect pathways were significantly under-represented in mice on the F/C– diet (Fig 6C). These results fit a view of 1C metabolism as a platform that primarily allocates resources for anabolic, biosynthetic pathways. Reducing the output of these pathways late in life is known to promote longevity in other settings.

Third, to gauge if and how a folate-limited diet late in life impacts gene expression, we measured transcript steady-state levels of liver tissue with RNAseq. We did not see extensive transcriptome remodeling in response to the folate-limited diet (Fig 7). Less than 5% of the 15,000–20,000 transcripts that entered the analysis changed in abundance significantly (>1.5-fold, $P$ < 0.05; see the Materials and Methods section and Fig 7A). Among the transcripts over-expressed in mice on the F/C– diet, there was no significant enrichment of any gene ontology biological process. On the other hand, among under-expressed transcripts, we found significant enrichment of transcripts encoding gene products involved in protein synthesis in both male (Fig 7B) and female animals (Fig 7C). The enrichment was stronger in males (>threefold) than in females (~twofold to threefold) but significant in both sexes (FDR < 0.05). These results agree with our metabolite analyses (see Fig 6C), arguing again that a folate-limited diet late in life induces a state of lower anabolism. As a readout of proliferative signaling, we next

looked at the levels of phosphorylation of ribosomal protein S6 (RPS6), which is an output of several kinase cascades, including the mTOR pathway (Meyuhas, 2015; Wu et al, 2022). Immunoblot analysis of liver tissue samples gathered at the time of euthanasia revealed variability in the detected values across individual mice. When examining the male mice, we observed that, on average, those fed the F/C– diet had approximately half the amount of phosphorylated RPS6 (P-RPS6) compared with those on the F/C+ diet. However, because of high variability in the measured values, the overall differences in P-RPS6 levels between the two dietary groups did not reach statistical significance (Fig S13; $P$ > 0.05, based on the Wilcoxon rank sum test). Similar results were obtained using antibodies that detect phosphorylation of 4EBP1 at Thr37,46, which is a known output of the mTOR pathway (Gingras et al, 1999) (Fig S14; $P$ > 0.05, based on the Wilcoxon rank sum test).

Lastly, we also measured IGF-1 levels in the sera of all animals at the endpoint of euthanasia using an ELISA-based assay (see the Materials and Methods section). The insulin/insulin-like growth factor pathway plays critical roles in growth control and longevity, mirroring the effects of the mTOR pathway. We found that female mice on the folate-limited diet had ~40% lower IGF-1 levels than their counterparts on the folate-replete diet (Fig S15; $P$ = 0.028, based on the Wilcoxon rank sum test). These data are again consistent with the notion that a folate-limited diet late in life may induce a state of lower cell growth and anabolism.

## Discussion

The results provide important insights into how adjusting folate intake in older adults may promote health at this later stage of life. Our data raise the exciting possibility that the amount of folate

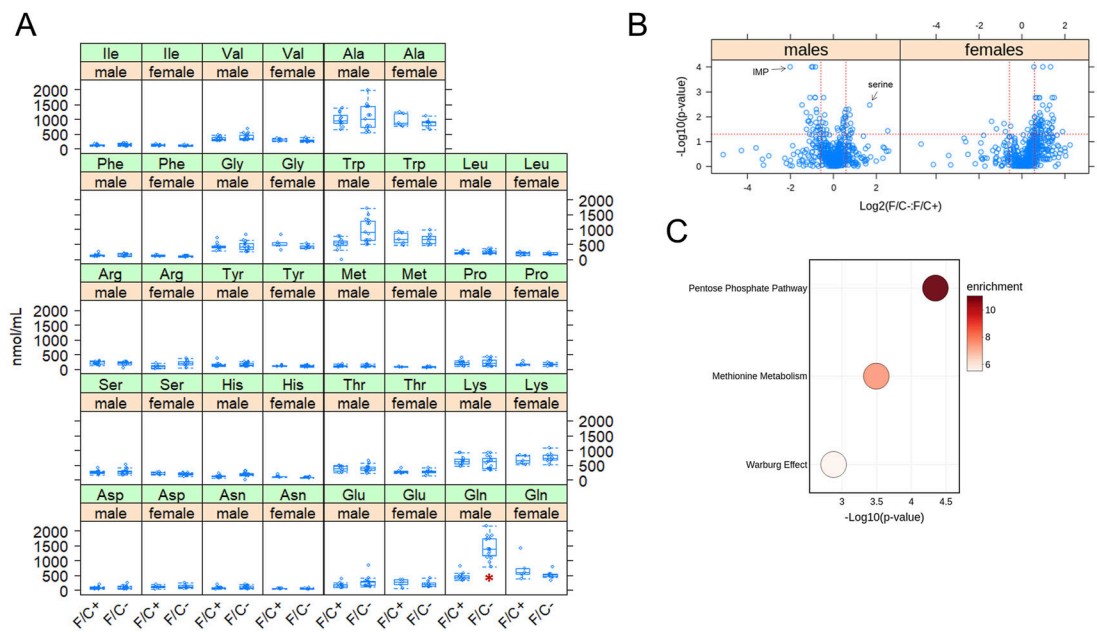

**Figure 6. Metabolite changes in mice placed on dietary folate restriction late in life.**
**(A)** Steady-state serum amino acid levels were measured at 120 wk of age using an HPLC-based assay (see the Materials and Methods section). The measured amounts (in nmol/ml) are on the y-axis. The different diets are on the x-axis for each amino acid, as indicated. The boxplots were drawn as in the previous figures. Differences in glutamine (Gln) levels in male mice were significant (marked with a red asterisk; $P = 2.6 \times 10^{-5}$, based on the Wilcoxon rank sum test). **(B)** Steady-state levels of primary and biogenic amine metabolites from liver tissue collected at 120 wk of age were measured by GC-TOF MS and HILIC-QTOF MS/MS, respectively. Changes in the indicated pairwise comparisons were identified from the magnitude of the difference (x-axis; $Log_2$-fold change) and statistical significance (y-axis; based on robust bootstrap ANOVA tests), as indicated by the red lines (see the Materials and Methods section). **(B, C)** Metabolite enrichment analysis based on the MetaboAnalyst platform (Chong et al, 2019) for male mice from the data in (B), for metabolites present at significantly lower levels under folate limitation ($P < 0.05$ and >1.5-fold change). The corresponding metabolic pathways are on the y-axis, and the $P$-values are on the x-axis. The size of each bubble on the chart reflects the relative number of "hit" metabolites in each pathway. The color of each bubble reflects the enrichment ratio, which is the number of hits within a metabolic pathway divided by the expected number of hits. Only pathways with enrichment >2 and FDR values < 0.05 are shown.
Source data are available for this figure.

needed to promote health changes at different stages of life and that reduced folate intake later in life might be beneficial.

What theoretical framework explains the proposed differential impacts of folate consumption during different stages of life? Antagonistic pleiotropy offers insight into these age-specific effects, suggesting that early-life advantages in fitness and reproduction come at a cost later (Williams, 1957). This theory, introduced by Williams in the '50s, proposes that "an individual cannot be exceptionally gifted with both youthful vigor and long life" (Williams, 1957). Antagonistic pleiotropy is now widely acknowledged in aging biology (Promislow, 2004; Austad & Hoffman, 2018), with the TOR pathway serving as a cardinal example (Blagosklonny, 2010).

Folate pathways and their role in healthspan may be analogous to the TOR pathway. 1C metabolism allocates resources for biosynthesis, providing precursors for nucleotides, proteins, and lipids (Fig 1A). 1C metabolism also adjusts the methylation and redox status of the cell (Ducker & Rabinowitz, 2017; Rosenzweig et al, 2018; Reina-Campos et al, 2020). 1C pathways are uniquely positioned to affect several established hallmarks of aging (Lopez-Otin et al, 2013). TOR orchestrates translation and various metabolic reactions, including 1C reactions (Ben-Sahra et al, 2016), to promote cell growth and proliferation (Barbet et al, 1996). Inhibiting the TOR pathway genetically or chemically inhibits cell division (Barbet et al,

1996) but promotes longevity (Johnson et al, 2013). Notably, such differential effects, predicted by antagonistic pleiotropy, could make folate interventions in aged individuals feasible and effective without impeding the benefits of high folate intake during early life.

Despite the evidence we outlined above from model organisms, is folate limitation even compatible with improved longevity in people? Genetic evidence supports this idea. Methylenetetrahydrofolate reductase (MTHFR) converts $CH_2$-THF to methyl-THF (see Fig 1A). A C677T mutation, leading to an Ala to Val substitution in the *MTHFR* gene, lowers MTHFR's activity by about half. Among French centenarians and nonagenarians with a family history of longevity, the prevalence of the *MTHFR* C677T allele was higher than in controls (<70 yr of age), approaching statistical significance ($P = 0.06$), and C677T carriers had extremely low-folate serum levels (Faure-Delanef et al, 1997). These observations suggest that individuals with lower serum folate levels *can* undoubtedly live longer.

Most animals, including humans, cannot synthesize folate. Folate and its various forms, including 5-methyltetrahydrofolate, the naturally occurring form absorbed through the intestinal mucosa, do not cross biological membranes. Hence, cells rely on receptors or transporters that bind folate and bring it inside cells. Dietary folate is absorbed primarily in the duodenum by the proton-coupled folate transporter (PCFT/SLC46A1) (Qiu et al, 2006; Visentin et al, 2014). Mice with a homozygous deletion of the *Pcft*

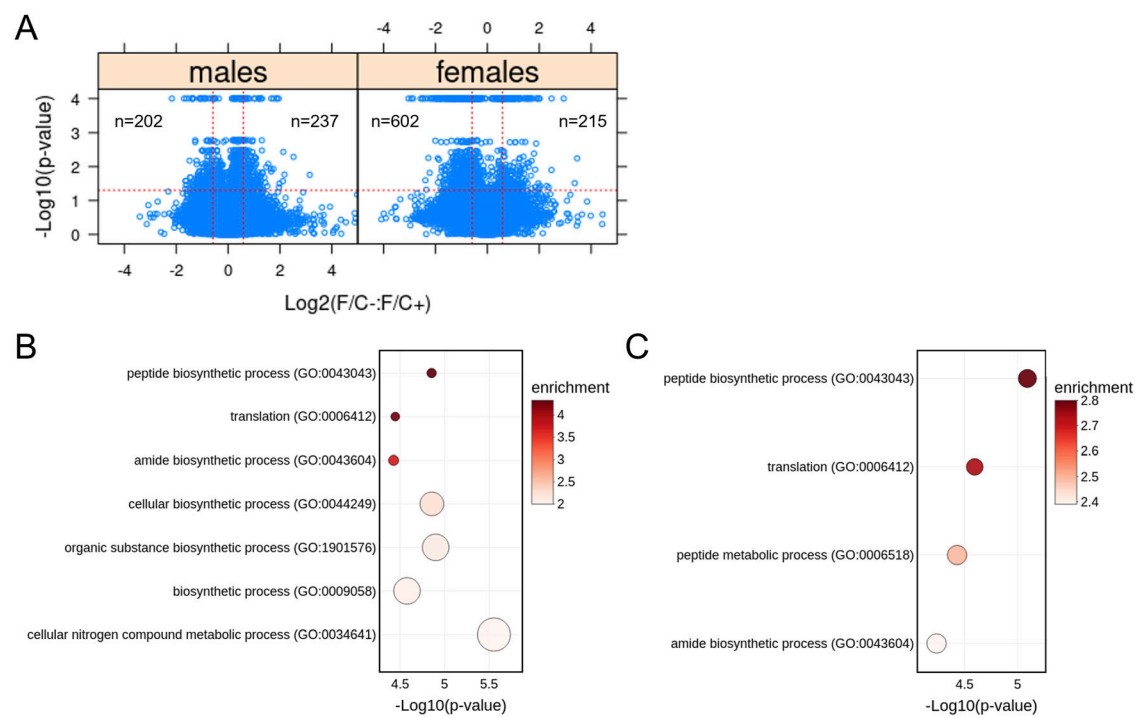

**Figure 7. Transcriptomic changes in mice placed on dietary folate restriction late in life point to reduced protein synthesis in both sexes.**
**(A)** Steady-state levels of mRNAs from liver tissue collected at 120 wk of age were measured by RNAseq. Changes in the indicated pairwise comparisons were identified from the magnitude of the difference (x-axis; $Log_2$-fold change) and statistical significance (y-axis; based on robust bootstrap ANOVA tests), as indicated by the red lines (see the Materials and Methods section). The number of transcripts whose levels changed within these thresholds is shown in each case. **(A, B)** Transcript enrichment analysis based on the PANTHER platform (Ashburner et al, 2000; Thomas et al, 2022; Gene Ontology Consortium et al, 2023) for male mice from the data in (A), for transcripts present at significantly lower levels under folate limitation ($P < 0.05$ and >1.5-fold change) that could be assigned to a specific gene ID (n = 165). The corresponding biological processes are on the y-axis, and the $P$-values are on the x-axis. The size of each bubble on the chart reflects the relative number of "hit" transcripts in each process. The color of each bubble reflects the enrichment ratio, which is the number of hits within a process divided by the expected number of hits. Only pathways with enrichment >2 and FDR values < 0.05 are shown. **(A, C)** Transcript enrichment analysis for female mice from the data in (A), for transcripts present at significantly lower levels under folate limitation ($P < 0.05$ and >1.5-fold change) that could be assigned to a specific gene ID (n = 490). The plots were drawn as in (B). Source data are available for this figure.

gene have defective erythropoiesis and fail to thrive, resembling the hereditary folate malabsorption syndrome in humans (Salojin et al, 2011). The heterozygous *Pcft* ± mice have significantly reduced (~50%) folate levels in the serum, liver, kidney, and spleen (Salojin et al, 2011). Remarkably, however, erythropoiesis, growth rate, and survival of these *Pcft* ± animals were indistinguishable from their wild-type counterparts (Salojin et al, 2011).

How could a folate-limited diet late in life improve healthspan? Our metabolomic and transcriptomic profiling experiments likely offer a possible and relatively straightforward answer, revealing signatures of down-regulated cellular anabolic processes in the folate-limited animals. Reduced anabolism is common among pro-longevity interventions, for example, when inhibiting the master regulator of cell growth, the TOR pathway (Johnson et al, 2013). We note that a down-regulation of anabolic signatures in the folate-limited animals was evident in both sexes (Fig 7), arguing that the changes we observed reflect broad and common outcomes of folate limitation. Another signature we observed, at least in males, was elevated serum glutamine levels (Fig 6A). In human studies, centenarians have elevated serum glutamine compared with 70-yr-olds (Montoliu et al, 2014). Furthermore, glutamine supplementation in elderly subjects may improve inflammatory responses, redox

balance, and exercise performance (Almeida et al, 2020). In contrast, low serum glutamine levels are associated with increased morbidity and mortality (Boelens et al, 2001), but it is unclear if and how the markedly elevated glutamine levels we observed promote longevity.

On the other hand, at least in model organisms, lowering the output of the IGF-1 pathway is known to promote longevity, and it was the first pathway shown to extend lifespan (Kenyon, 2011). Heterozygous female mice carrying a knockout allele of the IGF-1 receptor have ~30% longer lifespan (Holzenberger et al, 2003). Reduced serum levels of IGF-1 are associated with longer lifespan among a panel of inbred strains of mice (Yuan et al, 2009). The effects of IGF-1 signaling in people may also be a manifestation of antagonistic pleiotropy, with low levels being harmful early but beneficial later in life (Zhang et al, 2021). In this context, the lower IGF-1 levels we observed in female mice on the folate-limited diet are consistent with positive outcomes on longevity.

Lastly, we note that male and female mice on the folate-limited diet maintained their weight better as they aged than their counterparts on the folate-replete diet (Fig 2), likely through increased adiposity. A recent study identified these phenotypes as top predictors for positive effects on health and lifespan in

genetically diverse mice on various dietary restriction diets (Di Francesco et al, 2023 *Preprint*).

Although our mouse study suggests that restricting folate intake late in life is not harmful and may even be beneficial, we caution that there are significant limitations. For example, our data were from one inbred mouse strain and a relatively small sample size. Similar experiments in larger, genetically diverse populations are better suited for studying genome-by-diet interactions. Whereas exposure to the F/C− diet did not increase liver uracil DNA content, it is possible that more proliferative tissues, such as the colon, are more susceptible to decreased dietary folate availability (Chon et al, 2019). Moreover, our microbiome data (Fig 5) were limited in scope, relying exclusively on metagenomic pieces of evidence. The intestinal microbiome likely represents a significant source of folate and 1C metabolites. To properly investigate diet-host-microbiome relations, combining metagenomic and transcriptomic queries of the microbiome would be needed at various time points during the folate-related dietary interventions. Such approaches could more precisely identify which microbial species are present in what proportions and which microbial processes are critical in the responses to dietary folate limitation. Because mice engage in coprophagy, microbiome contributions to folate metabolism are bound to be substantial in this species. There are also significant differences in folate status between mice and people. For example, people have lower levels (~10–15 ng/ml) of serum folate than mice (Bailey et al, 2015), and the activity of dihydrofolate reductase, an enzyme essential for maintaining tetrahydrofolate pools—the folate form used in 1C reactions, maybe only 2% of that in rodents (Bailey & Ayling, 2009). Hence, mice are likely more refractory to a low-folate dietary intake.

Despite the above limitations, the experiments we described may have important implications for public health because the USA and many other countries mandate folate fortification of staple foods to harness the known benefits of folate supplementation during pregnancy and early stages of life. However, such policies affect people of all ages, and the impact of folate intake on broad healthspan parameters later in life is relatively unexplored. Lastly, that a folate-restricted diet likely leads to a less anabolic state offers new pharmacologic avenues to promote such a state, with known pro-longevity outcomes. Over the last several decades, many drugs have targeted folate-based metabolism, which could be further explored in longevity interventions.

# Materials and Methods

Materials and resources are listed on the Table 1.

### Replicative lifespan assays and cell size measurements in yeast

All the assays were performed in the standard *Saccharomyces cerevisiae* strain BY4742 (MAT*α his3Δ1 leu2Δ0 lys2Δ0 ura3Δ0*) on solid rich undefined media (YPD; 1% $^w/_v$ yeast extract, 2% $^w/_v$ peptone, 2% $^w/_v$ dextrose, 2% $^w/_v$ agar), as described previously (Steffen et al, 2009). Methotrexate was dissolved in DMSO and used

at the final concentrations shown in Fig 1B–D. The cell size measurements shown in Fig S1 were also performed in strain BY4742 in YPD. Briefly, overnight cultures were diluted to ~5 × 10⁵ cells/mL in fresh media, allowed to proliferate for 2–3 h at 30°C, and then methotrexate was added at the indicated final concentrations. 4–5 h later, cell size was measured with a Z2 Channelyzer, as described previously (Soma et al, 2014).

### Lifespan assays in worms

All the assays were performed at 20°C using *C. elegans* strain N2 and a bacterial strain (OP50) commonly used as food for the worms, as described previously (Sutphin & Kaeberlein, 2009). Briefly, the assays were performed on solid agar nematode growth media plates prepared fresh before each experiment. The bacterial lawn was exposed twice to a UV dose of 120 mJ/cm² using a UVC-515 Ultraviolet Multilinker (Ultra-Lum, Inc.). Streaking these UV-exposed bacteria to fresh LB agar plates (1% $^w/_v$ tryptone, 0.5% $^w/_v$ yeast extract, 1% $^w/_v$ sodium chloride) produced no visible colonies. Methotrexate, or the ATIC inhibitor, was first dissolved in DMSO and then added to the media used to prepare the plates after autoclaving (the media were kept in a 50°C water bath until the plates were poured). Mock-treated control plates contained only DMSO. At the start of each experiment, a sufficient number of eggs were collected from plates without any drugs and then placed on plates containing the indicated doses of each compound tested. After hatching and progression to the adult stage, animals were transferred to new plates (marked as the start of the lifespan assay) containing the drug tested and fluorodeoxyuridine (dissolved in water), added at 50 *μ*M to block hatching of new animals. The plates were scored at least every other day until all the worms died. If an animal responded to gentle touch, it was scored as alive, otherwise a death was recorded, and the animal was removed from the plate. Worms were transferred to fresh plates as needed (e.g., if there was evidence of microbial contamination, dryness/cracks on the agar surface, consumption of the bacterial lawn, or hatching of new animals that escaped the fluorodeoxyuridine block). The reported lifespans were compiled from several independent experiments performed over several months (9–10 mo for the methotrexate experiments and 4–5 mo for the ATIC inhibitor), each scored by multiple individuals (four to five persons per experiment). No experiments were excluded from the analysis.

### Mice

One cohort of female and male C57BL6/J mice, comprising 40 animals per sex, were purchased from Jackson Laboratories at 28 wk of age. All mice were housed on a 12-h light/dark cycle and kept at 20–22°C. Until 52 wk of age, all animals were fed the standard purified diet AIN-93M (Reeves et al, 1993), referred to as F/C+ throughout the text. At 52 wk of age, half the animals were switched to a modified AIN-93M diet lacking folate and choline, referred to as F/C− throughout the text until the termination of the study. Both diets were from Dyets Inc.; see Table 1. We note that when designing experiments to assess the consequences of folate limitation, it is common to control both folate and choline intake to ensure that

**Table 1. Key resources table.**

| Reagent type (species) or resource | Designation | Source or reference | Identifiers | Additional information |
|---|---|---|---|---|
| Strain (*S. cerevisiae*) | BY4742 | PMID: 10436161 | | MATα his3Δ1 leu2Δ0 lys2Δ0 ura3Δ0 |
| Strain (*C. elegans*) | N2 | PMID: 32550510 | RRID: SCR_014958 | |
| Strain (*M. Musculus*) | C57BL/6J | The Jackson Laboratory | RRID:IMSR_JAX: 000664 | |
| Antibody | Goat HRP-conjugated anti-rabbit secondary antibody | Thermo Fisher Scientific | 31466 | Used at 1:5,000 dilution |
| Antibody | Rabbit monoclonal antibody against phosphorylated ribosomal protein S6 at Ser235/236 | Cell Signaling | 4858 | Used at 1:5,000 dilution |
| Antibody | Rabbit polyclonal antibody against ribosomal protein S6 | Thermo Fisher Scientific | A300-557A | Used at 1:8,000 dilution |
| Antibody | Rabbit polyclonal antibody against phosphorylated ribosomal protein 4EBP1 at Thr37,46 | Thermo Fisher Scientific | PA5-17728 | Used at 1:1,500 dilution |
| Antibody | Rabbit monoclonal antibody against total 4EBP1 protein | Cell Signaling | 9644(53H11) | Used at 1:2,000 dilution |
| Commercial assay or kit | Mouse Cytokine/Chemokine 32-Plex Discovery Assay Array | Eve Technologies | MD32 | https://www.evetechnologies.com/product/mouse-cytokine-array-chemokine-array-32-plex/ |
| Commercial assay or kit | Quick-RNA Miniprep Plus Kit | Zymo Research | D7003 | |
| Commercial assay or kit | Mouse IGF-1 ELISA Kit | Abcam | ab100695 | |
| Chemical compound, drug | Methotrexate | Millipore Sigma | M9929 | |
| Chemical compound, drug | ATIC Dimerization Inhibitor | Millipore Sigma | 118490 | |
| Software, algorithm | R | https://www.r-project.org/ | | |
| Other | AIN-93M; F/C+ | Dyets, Inc | D110900 | |
| Other | Modified AIN-3M; F/C- | Dyets, Inc | D117879 | |
| Other | Yeast extract | Millipore Sigma | Y1625 | |
| Other | Peptone | Millipore Sigma | P5905 | |
| Other | Dextrose | Millipore Sigma | G7528 | |
| Other | Agar | Millipore Sigma | A1296 | |
| Other | Dimethyl sulfoxide; DMSO | Millipore Sigma | D1435 | |
| Other | Chlorhexidine diacetate; Nolvasan | Zoetis | | https://www.zoetisus.com/products/petcare/nolvasan-skin-and-wound-cleanser |
| Other | Blood sample tube with a clotting activator/gel | Sarsdedt | 41.1378.005 | |
| Other | Microvette CB 300 EDTA K2E, capillary blood collection tubes | Sarsdedt | 16.444.100 | |
| Other | DNA/RNA Shield solution | Zymo Research | R1100-50 | |
| Other | Proteinase K Solution | Thermo Fisher Scientific | 4333793 | |
| Other | DNAse I; TURBO DNase | Thermo Fisher Scientific | AM2239 | |

the observed effects are caused by the restriction of folate (Beaudin et al, 2011) because the presence of choline can mask the effects of folate deficiency. Choline can be oxidized to betaine, which provides methyl groups for converting homocysteine to methionine, independent of the folate cycle. Choline can also be incorporated into phosphatidylcholine, a major methyl "sink" in the cell, through the Kennedy pathway. Lastly, we did not use any antibiotics to interfere with the microbiome nor wire bottom cages to eliminate coprophagy. Wire bottom cages were used only in the metabolic chamber experiments.

Mice were housed in groups of no more than five animals per cage at the TAMU Comparative Medicine Program (CMP) facilities. There was no incident of aggressive males needing to be isolated to prevent fighting. All the animals were scored for various phenotypic metrics (see Fig 2A) at 42 wk of age before grouping into the different diets. Grouping was randomized based on lean mass values. In addition, partitioning into the two diet groups was balanced for all other metrics we evaluated so that for any given mouse in any given group, there were similar mice in the other diet group of the same sex.

The animals were inspected daily and treated for non-life-threatening conditions as directed by the veterinary staff of TAMU-CMP. The only treatment received was for dermatitis (topical solution thrice weekly, as needed). Each room contained sentinel animals housed in filtered cages. When cages were scheduled to be changed, a small amount of dirty bedding was collected from a (rotated) selected rack of cages and added to the fresh sentinel cage to ensure that the sentinel animals were uniformly exposed to any contaminant or pathogen that may be present within the colony. One sentinel animal from each group was sampled quarterly, and the serum was tested for common rodent pathogens. A veterinary pathologist performed microbiology, parasitology, and histology as deemed appropriate.

All animal protocols were approved by the TAMU Animal Care and Use Committee (IACUC 2020-0003; Reference#: 142420). TAMU has NIH/PHS Approved Animal Welfare Assurance (D16-00511 [A3893-01]), and TAMU-CMP is accredited by the Association for the Assessment and Accreditation of Laboratory Animal Care (AAALAC).

## Frailty Index

For all measurements, we followed established procedures (Whitehead et al, 2014). These measurements indicate age-associated deterioration of health and include the scoring of various integument, physical/musculoskeletal, ocular/nasal, digestive, and respiratory conditions. For example, integument scored alopecia, ruffled/matted coat, and piloerection. Physical/musculoskeletal conditions included tumors, distended abdomen, kyphosis, gait problems, tremors, and body weight. The ocular/nasal category covered cataracts, corneal opacity, eye discharge, and malocclusion. Diarrhea and rectal prolapse were the digestive phenotypes. Respiratory conditions observed were increased breathing rate and labored breathing. A score of 0 was assigned if no sign of frailty was observed and the animal was healthy for that phenotype. Moderate and severe phenotypes were scored as 0.5 and 1, respectively.

## Whole-body composition

The analysis was conducted on live, awake animals using a quantitative nuclear magnetic resonance machine (EchoMRI-100; EchoMRI LLC).

## Gait measurements

Assays were performed with the DigiGait system (Mouse Specifics, Inc.). Mouse paw pads were colored red with a non-toxic, washable marker and then placed on the transparent treadmill belt. The belt was started at 0° incline and 24 cm/sec speed. Video clips of 3–5 s were recorded once the mouse moved freely at full speed. Mice that did not move were retested once and marked as "did not move." Mice that did not move included a few that were likely physically too frail, but mostly were mice that used the bumper to allow the belt to drag under them without having to move. The video files were imported into the DigiGait Analysis program. After filtering out background noise, analysis graphs were generated and edited with the correct, connect, and exclude functions when necessary.

## Open field activity

Total time moving and time in the center of open field arenas were measured using the Noldus Ethovision video tracking software system (version 17.0.1630). Arenas were sectioned into a center, inner zone, outer zone, and a thigmotaxis area along the walls. Mice acclimated to the testing room for at least 15 min before being placed in the center of an arena and left undisturbed. For 30 min, the camera system tracked their center points.

## Novel object recognition

The assays were carried out using a layout consisting of eight gray-colored plexiglass arenas with cameras for video recording. Acquisition of video footage and analysis was performed with the Noldus Ethovision software. Objects were pre-tested weeks in advance to ensure that, on average, the mice did not prefer one object over the other. The two types of objects used for the experiment were constructed of plastic Trio blocks, using slightly different blocks and placing a metal spring on one set to make the two different configurations. Objects were thought to be "climb-proof," but specific mice still climbed on them. Time spent sitting on the objects was removed from the analysis, whereas quick "climb-throughs" of the objects were kept in the analysis. Eight identical sets of each of the object configurations were constructed.

The testing protocol involved the following steps. Mice in their home cages were placed in the testing room and allowed to acclimate for at least 0.5 h or until their activity slowed and they rested again. Mice (eight total, one in each arena) were then placed in empty testing arenas for a 10-min habituation phase. After habituation, mice were placed together back in their home cage for approximately 15 min. During that time, arenas were cleaned with a 2% ($^w/_v$) chlorhexidine diacetate solution (Nolvasan), and identical objects were placed in each arena. To avoid any bias from the order of the objects (familiar versus novel), half the arenas used one

object as the familiar object; half used the other. Mice were then placed back into their same arena, now containing the identical objects, and allowed to explore for 10 min, called the familiarization phase. The phase was recorded to ensure that mice explored the objects for at least 1 min. At the end of the familiarization, the mice were housed individually in separate holding cages for 15 min. During this time, arenas were cleaned with Nolvasan, and all objects were removed and cleaned with ethanol. New object configurations were set up in each arena, one object being the familiar object seen during the previous phase and one object being the new or novel object. The novel object placement varied between the four arenas to avoid any bias in spatial preference. After 15 min in the holding cages, mice were placed back in the same arena for 5 min of recording their exploration of the familiar and novel object.

Novel object exploration was analyzed using Noldus Ethovision Analysis software. Each trial was edited to ensure correct recognition of the head versus tail. Software arena settings were such that equal-sized regions around each object were drawn, and the software calculated the time of the mouse's nose in the area of interest around either the familiar or novel object. The amount of time the nose point was within 2 cm of either object was used to calculate a discrimination index (DI) (Lueptow, 2017). The DI here was defined as the time spent exploring the novel object minus the time spent exploring the familiar object, which was then divided by total exploration time. A discrimination index greater than or equal to 0.06 was considered a preference for the novel object.

## Echocardiography

Examinations were performed using the FujiFilm VisualSonics Vevo 3100 high-frequency ultrasound system. Mice were anesthetized using an isoflurane anesthesia chamber (SomnoSuite Mouse Anesthesia System). Once unconscious, mice were placed supine on a heated imaging platform (37°C) and provided a constant flow of isoflurane for the procedure. Limbs were carefully taped over electrodes containing electrode conductivity gel to maintain body position and monitor heart rate and respiration. Nair hair removing lotion was used to remove fur from the thoracic region of the animal. After removing the fur and cleaning with a damp cloth, ultrasound gel was added to the animal's chest for imaging. Mouse heart rates were kept between 285 and 500 bpm for imaging. Analysis was performed with the Vevo LAB software.

## Metabolic monitoring

We used PhenoMaster metabolic cages (TSE systems) to evaluate 32 individually housed mice (eight mice per sex per diet group). The system has small mammal gas calorimetry sensors to measure the animal's oxygen consumption and carbon dioxide production to calculate key metabolic parameters, including the RER. Energy expenditure, food intake, water consumption, body weight, and physical activity were simultaneously recorded over three consecutive days (72 h). Only data for the last 24 h of the experiment were used in the analysis to give the mice two days of acclimation to the new environment.

## Blood collections

Except for the terminal collection by cardiac puncture, all other blood collections were from the submandibular (facial) vein. The professional staff of TAMU-CMP did all the collections. For the terminal collection, each mouse was euthanized in a carbon dioxide chamber, and a cardiac puncture was performed soon after. Blood was placed in sample tubes with a clotting activator/gel and allowed to clot for at least 20 min. Tubes were then spun at 5,000$g$ for 5 min. The clarified serum was aliquoted and stored at –80°C.

## Complete blood counts

Samples were in Microvette CB 300 EDTA K2E capillary blood collection tubes. The tubes were inverted 10–15 times again before measuring with the Abaxis VetScan HM5 Color Hematology System.

## Inflammatory cytokines and chemokines

Serum samples were sent to Eve Technologies and measured with multiplex laser bead array technology (test MD32). The measurements for each mouse sample were performed in duplicate.

## Fecal microbiome analysis

There was no microbiome normalization between groups before the beginning of the experiment. Mouse fecal pellets were gathered by positioning the mice on a paper towel beneath an overturned glass beaker. A minimum of three fecal pellets from each animal were transferred into cryovials using sterile forceps. The samples were preserved at –80°C and shipped to Zymo Research, where they were processed and analyzed with the ZymoBIOMICS Shotgun Metagenomic Sequencing Service (Zymo Research).For DNA extraction, the ZymoBIOMICS-96 MagBead DNA Kit (Zymo Research) was used according to the manufacturer's instructions. Genomic DNA samples were profiled with shotgun metagenomic sequencing. Sequencing libraries were prepared with Illumina DNA Library Prep Kit (Illumina) with up to 500 ng DNA input following the manufacturer's protocol using unique dual-index 10 bp barcodes with Nextera adapters (Illumina). All libraries were pooled in equal abundance. The final pool was quantified using qPCR and TapeStation (Agilent Technologies). The final library was sequenced on the NovaSeq (Illumina) platform. The ZymoBIOMICS Microbial Community DNA Standard (Zymo Research) was used as a positive control for each library preparation. Negative controls (i.e., blank extraction control, blank library preparation control) were included to assess the level of bioburden carried by the wet-lab process.

Raw sequence reads were trimmed to remove low quality fractions and adapters with Trimmomatic-0.33 (Bolger et al, 2014): quality trimming by sliding window with 6 bp window size and a quality cutoff of 20, and reads with size lower than 70 bp were removed. Antimicrobial resistance and virulence factor gene identification was performed with the DIAMOND sequence aligner (Buchfink et al, 2015). Microbial composition was profiled with Centrifuge (Kim et al, 2016) using bacterial, viral, fungal, mouse, and human genome datasets. Strain-level abundance information was extracted from the Centrifuge outputs and further analyzed to

perform alpha- and beta-diversity analyses and biomarker discovery with LEfSe (Segata et al, 2011) with default settings ($P > 0.05$ and LDA effect size > 2).

## Serum folate assays

Liver and serum folate concentrations were measured by the *Lactobacillus casei* (*L. casei*) microbiological assay as previously described (Suh et al, 2000). *L. casei* growth was quantified at 595 nm on an Epoch Microplate Spectrophotometer (Biotek Instruments). Total folate measurements for the liver samples were normalized to protein concentration as measured using the Lowry-Bensadoun Assay (Bensadoun & Weinstein, 1976).

## Uracil in genomic DNA

Liver genomic DNA was isolated using the High Pure PCR Template Preparation Kit (Roche), followed by RNAse A treatment. As previously described, 2 μg of DNA were treated with uracil DNA glycosylase (Heyden et al, 2023). Samples were derivatized, and uracil levels were quantified using gas chromatography–mass spectrometry (Fiddler et al, 2021).

## Histopathology

After euthanasia, 10 organs were harvested from each mouse (see Fig S1). The samples were then fixed in 10% neutral buffered formalin at room temperature for 48 h and stored in 70% ethanol until processing. Tissue sections were processed using Leica ASP300 Tissue Processor for 4 h before being embedded in paraffin. The paraffin-embedded samples were sectioned at 5 μm, followed by hematoxylin and eosin staining, using a Leica HistoCore SPECTRA stainer. In a blinded manner, a board-certified veterinary pathologist (LG Adams) evaluated histologic sections of all tissues using brightfield microscopy, scoring tissue damage on a scale from 0 to 4.

## Metabolomic profiling

The untargeted, primary metabolite and biogenic amine analyses were performed at the West Coast Metabolomics Center at the University of California at Davis, according to their established mass spectrometry protocols. Extract preparation was also performed at the same facility from ~10 mg of liver tissue in each sample, provided frozen (at −80°C). To identify significant differences in the comparisons among the different groups, we used the robust bootstrap ANOVA via the t1waybt function of the WRS2 R language package (Mair & Wilcox, 2020). Detected species that could not be assigned to any compound were excluded from the analysis.

## Amino acid analysis

Serum samples were used for the PTH-based amino acid analyses (Heinrikson & Meredith, 1984) performed at the Texas A&M Protein Chemistry Facility. Statistical tests for significant differences between the different strains were performed as described above for the other metabolites.

## DNAge analysis

Liver samples (~15 mg) collected at euthanasia were placed in 0.75 ml of 1X DNA/RNA Shield solution (Zymo Research), shipped to Zymo Research, and processed with DNAge Service according to their established protocols. Briefly, after DNA extraction, the EZ DNA Methylation-Lightning Kit (Zymo Research) following the standard protocol was used for bisulfite conversion. Samples were enriched specifically for the sequencing of >1,000 age-associated gene loci using Simplified Whole-panel Amplification Reaction Method (SWARM), where specific CpGs are sequenced at minimum 1,000X coverage. Sequencing was run on an Illumina NovaSeq instrument. Sequences were identified by Illumina base calling software then aligned to the reference genome using Bismark. Methylation levels for each cytosine were calculated by dividing the number of reads reporting a "C" by the number of reads reporting a "C" or "T". The percentage of methylation for these specific sequences were used to assess DNA age according to Zymo Research's proprietary DNAge predictor which had been established using elastic net regression to determine the DNAge.

## RNA preparation from liver tissue

Liver samples were collected at euthanasia and stored at −80°C in 1X DNA/RNA Shield from Zymo Research. For RNA extraction, approximately 15 mg of the stored liver tissue was resuspended in 0.3 ml of fresh DNA/RNA Shield and processed using Zymo Research's Quick-RNA Miniprep Plus Kit. Extraction was performed according to the manufacturer's protocol with Proteinase K digestion for 4 h at room temperature, followed by a brief centrifugation to remove particulate debris. RNA lysis buffer from the kit was added to the clarified supernatant, and column purification with DNase I treatment was again performed according to the manufacturer's protocol. RNA was eluted in water and stored at −80°C.

## RNAseq

Initial Quality Control was performed for library construction to assess RNA concentration, OD ratios, and integrity. mRNA was isolated from 150 ng total RNA using a Nextflex Poly-A Selection kit (Perkin Elmer). cDNA libraries were prepared using a Nextflex Rapid Directional RNA 2.0 kit, miniaturized to 2/5 reaction volume, and automated on a Sciclone NGSx liquid handler. The 39 libraries were pooled by equal mass. The pool was sequenced on one Illumina NovaSeq S1 flowcell using the paired-end 2 × 50 bp recipe, which yielded 1,778 million raw clusters, with an average of 45 million clusters per sample. The sequencing raw data were mapped and quantified using the DRAGEN RNASEq pipeline with default parameters on a DRAGEN Bio-IT Platform with FPGA acceleration. 11% of the bases were trimmed based on quality (minimum quality of 24) and adapters (Stringency of 5). The resulting reads had an average mapping rate of 85% across the samples. The reference genome used was Mouse Build 39 Jun 2020, GCA_000001635.9. Transcript per million values were used in all downstream analyses.

## Immunoblots

In all cases we used liver tissue samples collected at euthanasia. ~20 mg of frozen liver tissue stored in −80°C was placed into 0.1 ml cold RIPA buffer (150 mM NaCl, 1.0% IGEPAL CA-630, 0.5% sodium deoxycholate, 0.1% SDS, 50 mM Tris, pH 8.0) containing protease and phosphatase inhibitors. Tissues were manually homogenized with a disposable pestle and microcentrifuge tube. Sample volume was brought to 0.4 ml with additional cold RIPA buffer and the lysates were centrifuged at 10,000$g$. Clarified lysate was transferred to a new tube and an aliquot removed for immunoblotting.

For the RPS6 immunoblots, two gels were loaded for each set of samples, one to detect total RPS6, the other to be used for detection of phosphorylated RPS6. RPS6 phosphorylation was detected by a specific rabbit monoclonal antibody against phosphorylated Ser235/236 of the human RPS6 protein, followed by an HRP-conjugated anti-rabbit secondary antibody (see Table 1). Total amounts of RPS6 were detected with a rabbit anti-RPS6 polyclonal antibody (see Table 1). The phosphorylated RPS6 signal from each mouse sample was divided by the total RPS6 signal in that sample. To account for spurious differences arising among the different gels, on each gel we run samples from the same sex, but from the two diet groups. Then, each P-RPS6:RPS6 ratio was divided by the average of these ratios from all the samples on that gel (i.e., from both diet groups), and these are the values used to generate Fig S13. The raw immunoblots are in Fig S13—source data.

For the 4EBP1 immunoblots, the samples were prepared and run on SDS–PAGE gels as above. The antibodies used to detect phosphorylated 4EBP1 (at T37,46) and total 4EBP1 are described in the Table 1. We verified that the signal from the phospho-specific antibody was sensitive to treatment of $\lambda$-phosphatase (not shown). The quantification of the signals is shown in Fig S14, and the raw immunoblots are in Fig S14—source data.

## ELISAs

We used commercially available kits to measure serum levels of IGF-1 (see Table 1) according to the manufacturer's instructions. Measurements were taken using a Promega Glomax-Multi Detection System.

## Statistical analysis

Data were analyzed using the latest version of the R language. The corresponding R packages and tests are described in the text in each case. Briefly, longitudinal measurements were evaluated with mixed effects models using the lme4 and lmer packages. A typical function had the following syntax: lmer("measurements" ~ diet + time + sex + [1|ID], data = "dataset"). It fits a model where "measurements" is the response variable, and "diet," "time", and "sex" are the fixed effects. "ID" is an individual mouse, considered as a random effect to account for the non-independence of measurements within the same subject. This was a typical setup for a longitudinal evaluation where multiple measurements were taken from the same subjects over time. For survival analyses, we used the survival package. For group comparisons of non-longitudinal data, we used the non-parametric Wilcoxon rank sum test,

implemented with the base "wilcox.test" R function, or the robust bootstrap ANOVA, implemented with the "t1waybt" function of the WRS2 package. All the replicates in every experiment shown were biological ones. The number of biological replicates analyzed in each case is indicated in the text and the corresponding figures. No data or outliers were excluded from any analysis.

## Data Availability

All the RNAseq data have been deposited to the Gene Expression Omnibus (GEO; Accession GSE245438).

## Supplementary Information

## Acknowledgements

We acknowledge the use of several Texas A&M University core facilities, including: The Rodent Preclinical Phenotyping Core, the mouse housing facilities of the Comparative Medicine Program (CMP), the Protein Chemistry Laboratory of the Department of Biochemistry and Biophysics, and the Agrilife Genomics and Bioinformatics Service. We thank Jean-Philippe Pellois (Texas A&M University) for making available his plate reader for ELISA-based measurements. This work was supported by a Texas A&M T3 award to M Polymenis, HL Andrews-Polymenis, and R Sitcheran. NIH grants support M Polymenis (R01GM123139) and LG Adams (R01HL148153-01A1 and R21AI153879).

### Author Contributions

HM Blank: data curation, formal analysis, investigation, visualization, methodology, and writing—review and editing.
SE Hammer: investigation.
L Boatright: investigation and writing—review and editing.
C Roberts: investigation and writing—review and editing.
KE Heyden: investigation and writing—review and editing.
A Nagarajan: investigation and writing—review and editing.
M Tsuchiya: investigation and writing—review and editing.
M Brun: formal analysis, investigation, methodology, and writing—review and editing.
CD Johnson: resources, methodology, and writing—review and editing.
PJ Stover: methodology and writing—review and editing.
R Sitcheran: resources, funding acquisition, methodology, and writing—review and editing.
BK Kennedy: resources, supervision, methodology, and writing—review and editing.
LG Adams: resources, supervision, funding acquisition, investigation, methodology, and writing—review and editing.
M Kaeberlein: resources, formal analysis, supervision, methodology, and writing—review and editing.
MS Field: resources, supervision, methodology, and writing—review and editing.

DW Threadgill: supervision, methodology, and writing—review and editing.

HL Andrews-Polymenis: resources, formal analysis, supervision, funding acquisition, methodology, and writing—review and editing.

M Polymenis: conceptualization, resources, data curation, formal analysis, supervision, funding acquisition, investigation, visualization, methodology, project administration, and writing—original draft, review, and editing.

## Conflict of Interest Statement

The authors declare that they have no conflict of interest.

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
