## [Reviewer comments · Life Science Alliance]

Life Science Alliance

Late-life dietary folate restriction reduces biosynthesis without compromising healthspan in mice

Heidi Blank, Staci Hammer, Laurel Boatright, Courtney Roberts, Katarina Heyden, Aravindh Nagarajan, Mitsuhiro Tsuchiya, Marcel Brun, Charles Johnson, Patrick Stover, Raquel Sitcheran, Brian Kennedy, L. Garry Adams, Matt Kaeberlein, Martha Field, David Threadgill, Helene Andrews-Polymeris, and Michael Polymeris

DOI: <https://doi.org/10.26508/lsa.202402868>

Corresponding author(s): Michael Polymeris, Texas A&M University

Review Timeline:

Submission Date:	2024-06-05
Editorial Decision:	2024-06-28
Revision Received:	2024-06-29
Accepted:	2024-07-01

Transaction Report:

Please note that the manuscript was reviewed at *Review Commons* and these reports were taken into account in the decision-making process at *Life Science Alliance*.

Review
COMMONS

Reviews

Review #1

****Summary:**** the work presented by the authors detail how pharmacological inhibition of the rate limiting one carbon metabolic enzyme DHFR by the drug methotrexate increases the lifespan of yeast and worms. Furthermore, placing aged mice on dietary folate and choline restriction potentially enhanced metabolic plasticity but did not significantly increase lifespan with sex specific differences observed.

The findings in this manuscript are very interesting and important to our understanding of the conserved mechanisms that regulate longevity through one carbon metabolism. This is especially significant in light of the current folate intake and supplementation in the adult human population. The manuscript, however, requires major revisions. Please see comments below for details.

****Major comments:****

1. The overall tone in this manuscript is colloquial and conversational in nature. A third person academic style and tone, while avoiding the use of subjective descriptive terms would improve the quality of this text. Using terms such as "appeared less diverse", "results are remarkable ...strikingly more pronounced", "possibly positive outcomes", "appear younger...for unknown reasons", "little Uracil", "tended to be higher", "roughly proportional", "slightly higher", "as a rough readout", and many other examples from the text should not be used in a scientific manuscript. The language should be academic, scientific, precise, and non-ambiguous. A thorough revision of the manuscript with substantial changes to the language and tone is necessary prior to publication.

2. In the results section, we find multiple instances where the results are interpreted and extensively discussed. This should be reserved for the discussion section. The results section should be used to simply report the findings in a detailed manner.

3. The materials and methods section is severely lacking in details in some areas. For example, no details were provided regarding how the worm lifespans were conducted and previous work of collaborators were referenced instead. Important details such as worm numbers, biological and technical replicates, solid agar vs liquid culture, temperature, use of FUDR, antibiotics, transfer frequency, methods of scoring, etc... are lacking. Other details such as the preparation of the plates (Was MTX incorporated into the agar, seeded with the bacterial lawn, or liquid culture was used), storage conditions, age of the plates when lifespan started, how was the UV killing of the lawn verified etc...

many other methods subsections lack crucial details. Please carefully review the methodology and include sufficient pertinent details.

4. In the worms, interventions that impact germline proliferation can extend lifespan. Methotrexate is known to impact germline proliferation and can lead to toxic developmental effects and germline arrest. Was fecundity impacted by methotrexate using the dosages found to extend lifespan?

5. The authors stated that UV killed bacteria was used in the worm experiments but did not provide the reasoning for it. Virk had concluded that reduced bacterial pathogenicity is responsible for the lifespan extension and not the worm's OCM. How does your work agree with or refute these previous findings?

6. The authors state that AICAR (100 uM administration to the worms (no experimental details were given) increases their lifespan and concluded that this is proof that manipulation of 1C metabolism promotes longevity. There are 2 concerns here; first, AMPK activation leads to inhibition of TOR and that has been shown to promote longevity in multiple models. While we agree that a significant crosstalk between TOR and OCM exists, this experiment does not necessarily contribute to the argument that the authors are making. Second, it has been established by multiple groups that inhibition (RNAi and pharmacological) of DHFR1, TYMS1, SAMS1 and possibly other OCM enzymes leads to lifespan extension in worms. These findings provide stronger evidence that OCM regulates organismal longevity.

7. In the mouse study, the authors do not provide a rationale on why a folate and choline deficient diet was adopted as opposed to only a folate deficient diet. Additionally, we assume that the diets did not contain antibiotics (succinyl sulfathiazole) to reduce microbiome folate production since it was not mentioned. Where were bottom cages used to eliminate coprophagy? Were there any significant differences between male and female serum folate levels that could have contributed to the endpoints. Was only a subset of samples assayed for total folate? (fig 2b shows a possible n

of 6 per group?). If no antibiotics and no wire bottom cages were used, mice can maintain adequate folate levels from coprophagy without developing signs of anemia. Please discuss these details as it helps clarify the conditions used.

8. There are instances in the results section where statements were made implying that there are differences observed "slightly higher", "negative association" when it is not statistically significant. There can be either statistically significant differences/correlation or not. please be precise in your wording.

9. Graying was observed less significantly in the F/C- group according to the authors. However, no quantitative assessment was made, and it is merely observational. Inference to inhibition of mTOR was made, but mTOR protein and phosphorylation levels were not performed. The authors did perform western blotting on ribosomal S6 protein, however no assessment of the downstream mTOR targets P70S6k1 and 4EBP are shown.

10. Can the change in RER in F/C- mice compared to controls be explained by the increased adiposity in these animals?

11. How was the microbiome normalized between groups prior to the beginning of the experiment? (fecal slurry gavage, bedding exchange, cohabitation, none of the above?). There is no mention of this crucial step in the materials and methods section. Furthermore, additional details regarding the microbiome analysis are required (analysis pipeline, read depth, denoising, software, data processing, PCA analysis, etc...). it is not sufficient to state that Zymo performed the analysis. What is an "easily distinguishable gut microbiome" and "appeared less diverse"? a two-dimensional plot using two principal components would be more suitable for image 5A and allow for better visualization of the clustering of the groups. Since the authors suggest that the microbiome could be a source of 1C metabolites (including natural folate), it is important to clarify if coprophagy is involved.

12. How are inflammatory cytokines and marker levels linked to reduced anabolism and immune function in non-challenged animals?

13. When discussing the epigenetic analysis, the authors state "no changes in the DNA methylation from liver samples.." and "groups appear younger than expected". Please clarify these statements. Additional details are needed regarding the analysis performed and the choice of methylated loci and methods. Please reference the epigenetic clock or model that was used and if was developed for the same strain and sub-strain of mice. Is it using a modified "Hovarth" mouse DNA age epigenetic clock? If so, provide the necessary details and a possible explanation for the discrepancy other than "unknown reasons"

14. Regarding Uracil misincorporation, the liver contains significant stores of folate as it is the main hub for several critical OCM reactions (Phospholipid methylation is a major one). Earlier studies used antibiotics with or without coprophagy prevention measures to induce a state of folate depletion to induce uracil incorporation in various tissues of rodent models. There is some controversy whether dietary folic acid restriction/methyl donor restriction alone will lead to uracil misincorporation when there is no apparent depletion or anemia. Please discuss your specific experimental procedures and how it agrees or disagrees with the published literature.

15. The section discussing RPS6 needs to be rewritten and it is difficult to understand. Furthermore, as stated previously, considering phosphorylation of mTOR and its downstream targets 4EBP and S6K1 will give a clear indication of proliferative signaling. Additionally, these pathways are impacted by feeding status, diurnal cycles, and sex. Were these factors controlled prior to sacrifice? Where the animals sacrificed at the same time? In a fed or unfed state?

16. The western blots provided in supplementary files show uneven protein loading across lanes (ponceau stain). No loading control is shown such as B-actin. A separate blot is used for total and phosphorylated proteins as opposed to gently stripping the membrane of the phosphorylated blot and re-incubating with the antibody for total. While normalizing phosphorylated to total protein levels will eliminate some of the variability in the author's method. The uneven loading may introduce errors in the calculated ratios.

17. While the authors referenced older studies utilizing low dose methotrexate on rodents and provided a composite lifespan based on these findings, why was dietary folate and choline restriction used instead of a low dose methotrexate in mice in the current study? Please provide a rationale for this approach.

****Minor comments:****

1. While the authors make compelling arguments that lower folate intake later in life may promote healthy aging, an important consideration in the human population that a considerable percentage of older individuals may be consuming an excessive amount of folate due to the combination of fortification and voluntary supplementation. An alternate hypothesis that could apply to humans and lab models is that the existing levels of exposure to folate/folic acid may be accelerating the aging process and promoting disease in later life.
2. The common C57BL/6j is being referred to as the "long lived strain". Is this relative to mice in wild conditions? There are many transgenic C57bl/6 strains that live considerably longer. Please clarify if this is meant to describe the aged mice used in the experimental process.
3. While the authors state early in the manuscript that longevity was not a measured outcome in the mouse study, the manuscript contains statements discussing animal survival in the results and survival curves (figure 2). This gives the impression that the study was planned as a survival analysis initially and since no difference was observed between the experimental groups during the earlier stages, the secondary endpoints of health span analysis were adopted. Either approach does not detract from the significance of the study's findings. Further clarity on the approach would be beneficial to the readers.
4. For yeast culture conditions, what are the folate sources and content? Is there added folic acid similar to cell culture conditions where supraphysiological concentrations are used in standard mediums (RPMI and DMEM).
5. In the metabolism section, the authors make statements such as "the differences were minimal" , "probably were due..", "minimal effects", "apparent increase", "tended to be", "little uracil" etc.. please refrain from using subjective language and use precise scientific terms.
6. Figure 2-c, there is a typo, Weeks not months

**** Referees cross-commenting****

while we generally agree with the other reviewer's concerns, we find that reviewer 3 rejection of the authors conclusion without considering the evidence presented in the context of what is currently known in the field potentially limiting. Multiple groups have shown that manipulation of OCM enzymes (DHFR, TYMS, SAMS) can extend lifespan in worms. the recent report Antebi's group (Annibal et al. Nature Com, 2021) provides strong evidence that OCM is central to longevity regulation in worms and mice and that folate intake can interact with and modulate organismal longevity. while this manuscript findings are not conclusive, I think it is premature to dismiss it completely. perhaps the alternative is to discuss the limitations of this approach and interpret the results (or the lack of significant differences) in order to help guide future research into this important subject. generalizing rodent results to human is always going to be a limiting factor in this type of work. Mice have significantly higher circulating folate. additionally, DHFR activity (the rate limiting enzyme in folate OCM) in rodents can be up to 100 times higher than its human equivalent. another consideration is that mice, similar to other rodents, engage in coprophagy, thereby recycling and supplementing bacterially produced folate in the absence of antibiotics in the diet. Therefore, mice placed of dietary folate restriction in the absence of antibiotics do not develop signs of anemia or deficiency. Therefore, it could be argued that there is no loss of nutrients in mice in this scenario and that supplementation at the arbitrarily recommended level of synthetic folic acid (2mg/kg day) or higher could impact health and aging. Similarly , in humans excess folate intake has been controversially associated with a number of deleterious health effects. It is important not to dismiss these reports and encourage further research into this subject that impacts a significant percentage of the human population due to the widespread use of supplements.

A major strength of this study is that the authors show that manipulation of OCM either through pharmacological inhibition or dietary restriction can impact organismal longevity in a conserved manner across species from yeast to worms and mammals. These findings provide compelling evidence that folate intake and metabolism in humans should be rigorously researched as potential regulator of aging. These findings complement and agree with a recent report by Antebi's group (Annibal et al. Nature Com, 2021) highlighting that long-lived worm and mice strains exhibit similar metabolic regulation of one carbon metabolism. In the same report low levels of folate supplementation partially or completely abrogated the lifespan extension in some models. This study provides additional evidence that restricting OCM through drugs or dietary restriction can significantly impact healthspan and lifespan. Additionally, it raises the question whether excessive folate intake in aged adults may have potentially deleterious effects on health

and longevity. The limitations of this study can be seen in the overall lack of significant impact of the dietary intervention on the health metrics that were measured in mice. The study does not provide strong evidence that restricting folate and choline intake will produce favorable effects on health. Similarly, no significant impact on mice lifespan was observed based on the partial lifespan analysis. Further clarity is needed regarding the experimental procedures and methods used. The study, nonetheless, is an important step towards investigating the role of folate and OCM in regulating mammalian healthspan and lifespan. Future studies can expand on these findings and investigate whether OCM interventions that are started in early life can produce significant and measurable effects on longevity and health in mammals. The findings here provide a conceptual and incremental advance in our understanding of these complex interactions.

These findings are important to the research communities especially in the areas of longevity, metabolism, and nutrition.

Review #2

****Summary:**** In this manuscript they investigate whether disruption of the folate cycle can slow ageing/improve health in yeast, worms and mice. There are a few experiments in yeast and *C. elegans* but the rest is a meta analysis of some old data on folate-deprived mice and their own study of mice on a diet with and without folic acid and choline. They find that various interventions of the folate cycle extend lifespan in yeast and worms, that the old study suggest mice live longer without folic acid supplementation and that there is no change to healthspan with mice without folic acid and choline in the diet late in life and that these mice show some positive benefits. Analysis of the microbiome and the transcriptomics suggest small changes to the microbiota and changes in gene expression. Overall the authors conclude that biosynthetic processes have been inhibited without negative effects on healthspan.

****Major comments****

1. The two worm lifespan experiments in Fig 1 show very different controls despite the methods stating that the conditions were the same. Controls can vary from one experiment to another but the difference is striking. It would be good to have supplementary data about the number of repeats and other data about these experiments.
2. The diet lack folic acid and choline yet the conclusions are only about folate. The choline aspect of the diet needs to be acknowledged as a potential factor.
3. The authors argue that the effects on the mice are not mediated effects on the diet by the microbiome because there is not a statistical effect on diversity. However they do show a clear difference at the metagenomic level that fits with a metabolic difference. It also ignores work in *C. elegans* showing that inhibition of bacterial folate synthesis increases lifespan, not by decreasing folate supply but because lowered bacterial folate prevents an age-accelerating activity in the bacteria (Virk et al 2016). It has also been shown that a breakdown product of folic acid can be taken up by bacteria and influence ageing (Maynard et al 2018). I do not think the evidence is strong enough to discounted that the changes seen in the mice are not mediated by microbes.

****Minor comments****

1. It had been shown a long time ago that *sams-1* mutants in *C. elegans* extend lifespan. MTX is likely to influence SAMS levels. This point needs to be mentioned.
2. Page - 6 "folate accelerates worm aging". This statement is not correct and is not what Virk et al 2016 suggests.
3. Page 7. "at 100µM, a dose similar to the one used in mice with metabolic syndrome (Asby et al., 2015)." It's not valid to compare the concentration of a drug in the media in a *C. elegans* experiment to a dose given to mice.

**** Referees cross-commenting****

I would like to add that it is important to consider whether there are in fact negative effects of folic acid given in later life and this is one of the only studies that addresses this question in a mammalian model, and thus needs to be reported, once the issues raised have been addressed.

The main strength of this manuscript is that it examines the effect of mice given a folate and choline deficient diet late in life and finds mostly positive effects. This finding challenges the dogma that folate

Review #3

Blank/Polymenis and colleagues explore how reduced folate metabolism impacts aging. While folate supplementation is known to benefit the development and health of young people, little is known about the impact of this substrate at advanced ages. The paper consists of two parts: 1) blocking folate metabolism in yeast and *C. elegans* while measuring lifespan (reproductive or age of death); 2) measuring a vast array of traits in mice where folate (and choline) is removed from the diet starting at age 1 year. The second approach is most central to the paper's theme, and the authors conclude their 'data raise the exciting possibility that ... reduced folate intake later in life might be beneficial.' However, I do not accept this conclusion. Instead, the overwhelming fact is that there were no changes in any phenotype due to the absence of F/C in the older animals. Loss of this nutrient is neutral, although perhaps bad for the kidney. In my view, the authors misinterpret their very basic results: loss of dietary folate has no impact on aged mice (one strain, at that). And there is no way to generalize this simple conclusion to humans. There are other issues throughout the work that need to be addressed but given weakness on its key argument, I will not elaborate these points.

Blank/Polymenis and colleagues explore how reduced folate metabolism impacts aging. While folate supplementation is known to benefit the development and health of young people, little is known about the impact of this substrate at advanced ages.

RESPONSE TO REVIEWS

We thank all the reviewers for their comments and suggestions. Our point-by-point response is shown below, in **bold**.

Reviewer #1 (Evidence, reproducibility and clarity (Required)):

Summary: the work presented by the authors detail how pharmacological inhibition of the rate limiting one carbon metabolic enzyme DHFR by the drug methotrexate increases the lifespan of yeast and worms. Furthermore, placing aged mice on dietary folate and choline restriction potentially enhanced metabolic plasticity but did not significantly increase lifespan with sex specific differences observed.

The findings in this manuscript are very interesting and important to our understanding of the conserved mechanisms that regulate longevity through one carbon metabolism. This is especially significant in light of the current folate intake and supplementation in the adult human population. The manuscript, however, requires major revisions. Please see comments below for details.

Major comments:

1. The overall tone in this manuscript is colloquial and conversational in nature. A third person academic style and tone, while avoiding the use of subjective descriptive terms would improve the quality of this text. Using terms such as "appeared less diverse", "results are remarkable ...strikingly more pronounced", "possibly positive outcomes" , "appear younger...for unknown reasons", "little Uracil", "tended to be higher", "roughly proportional", "slightly higher", "as a rough readout", and many other examples from the text should not be used in a scientific manuscript. The language should be academic, scientific, precise, and non-ambiguous. A thorough revision of the manuscript with substantial changes to the language and tone is necessary prior to publication.

RESPONSE: Thank you for your feedback on the manuscript's tone. We revised most of the expressions mentioned by the reviewer. We note, however, that these phrases were used along with numbers and statistics. Hence, there was no lack of specifics, and readers could quickly evaluate the conclusions. We strive for a balance between scientific rigor and readability to maintain accessibility for a diverse audience.

2. In the results section, we find multiple instances where the results are interpreted and extensively discussed. This should be reserved for the discussion section. The results section should be used to simply report the findings in a detailed manner.

RESPONSE: We appreciate the suggestion on the integration of interpretation within the Results section. Upon review, we have clarified the presentation of our findings, ensuring a more distinct separation from interpretive commentary. Brief explanations remain to aid the reader's comprehension in light of the complex data, aiming to keep the flow and

coherence of the manuscript and prevent overextension of the Discussion section (already ~1,300 words long). We welcome specific suggestions for further refinement.

3. The materials and methods section is severely lacking in details in some areas. For example, no details were provided regarding how the worm lifespans were conducted and previous work of collaborators were referenced instead. Important details such as worm numbers, biological and technical replicates, solid agar vs liquid culture, temperature, use of FUdR, antibiotics, transfer frequency, methods of scoring, etc... are lacking. Other details such as the preparation of the plates (Was MTX incorporated into the agar, seeded with the bacterial lawn, or liquid culture was used), storage conditions, age of the plates when lifespan started, how was the UV killing of the lawn verified etc...

many other methods subsections lack crucial details. Please carefully review the methodology and include sufficient pertinent details.

RESPONSE: The number of worms assayed in each case were shown in each figure, as described in the legend. We now also added all the information requested by the reviewer in the methods section. The text now reads:

“Briefly, the assays were done on solid agar nematode growth media (NGM) plates prepared fresh before each experiment. The bacterial lawn was exposed twice to a UV dose of 120mJ/cm² using a UVC-515 Ultraviolet Multilinker (Ultra-Lum, Inc.). Streaking these UV-exposed bacteria to fresh LB agar plates (1% w/v tryptone, 0.5% w/v yeast extract, 1% w/v sodium chloride) produced no visible colonies. Methotrexate, or the ATIC inhibitor, was first dissolved in dimethyl sulfoxide (DMSO) and then added to the media used to prepare the plates after autoclaving (the media were kept in a 50°C water bath until the plates were poured). Mock-treated control plates contained only DMSO. At the start of each experiment, a sufficient number of eggs were collected from plates without any drugs and then placed on plates containing the indicated doses of each compound tested. After hatching and progression to the adult stage, animals were transferred to new plates (marked as the start of the lifespan assay) containing the drug tested and fluorodeoxyuridine (FUdR; dissolved in water), added at 50µM to block hatching of new animals. The plates were scored at least every other day until all the worms died. If an animal responded to gentle touch, it was scored as alive, otherwise a death was recorded, and the animal was removed from the plate. Worms were transferred to fresh plates as needed (e.g., if there was evidence of microbial contamination, dryness/cracks on the agar surface, consumption of the bacterial lawn, or hatching of new animals that escaped the FUdR block). The reported lifespans were compiled from several independent experiments done over several months (9-10 months for the methotrexate experiments and 4-5 months for the ATIC inhibitor), each scored by multiple individuals (4-5 persons per experiment). No experiments were excluded from the analysis.”

4. In the worms, interventions that impact germline proliferation can extend lifespan. Methotrexate is known to impact germline proliferation and can lead to toxic developmental

effects and germline arrest. Was fecundity impacted by methotrexate using the dosages found to extend lifespan?

RESPONSE: We did not score fecundity in our experiments.

5. The authors stated that UV killed bacteria was used in the worm experiments but did not provide the reasoning for it. Virk had concluded that reduced bacterial pathogenicity is responsible for the lifespan extension and not the worm's OCM. How does your work agree with or refute these previous findings?

RESPONSE: The dose of methotrexate used by Virk et al was very high, so it is difficult to directly compare it to our experiment. Nonetheless, we do not think there is any contradiction. We added the following in the text to clarify this point:

“At higher doses (10-100 μ M), methotrexate did not extend lifespan (not shown), in agreement with (Virk et al., 2016), who treated adult animals with a very high dose of methotrexate (220 μ M). We also note that the bacteria used to feed the worms in our experiments were killed by ultraviolet radiation to exclude any impacts from bacterial folate metabolism, which is known to affect worm lifespan (Virk et al., 2016, 2012).”

6. The authors state that AICAR (100 uM administration to the worms (no experimental details were given) increases their lifespan and concluded that this is proof that manipulation of 1C metabolism promotes longevity. There are 2 concerns here; first, AMPK activation leads to inhibition of TOR and that has been shown to promote longevity in multiple models. While we agree that a significant crosstalk between TOR and OCM exists, this experiment does not necessarily contribute to the argument that the authors are making. Second, it has been established by multiple groups that inhibition (RNAi and pharmacological) of DHFR1, TYMS1, SAMS1 and possibly other OCM enzymes leads to lifespan extension in worms. These findings provide stronger evidence that OCM regulates organismal longevity.

RESPONSE: We acknowledged prior research on lifespan extension and do not claim our use of the ATIC inhibitor as the first evidence of 1C metabolism's impact on longevity. Rather, our findings complement existing studies from us and several other groups (including the examples mentioned by the reviewer, which we had cited) by introducing novel evidence of lifespan increase through this specific inhibitor in *C. elegans*. Please also note that we added a detailed description of the experiment in the Methods, as suggested in a previous comment.

7. In the mouse study, the authors do not provide a rationale on why a folate and choline deficient diet was adopted as opposed to only a folate deficient diet. Additionally, we assume that the diets did not contain antibiotics (succinyl sulfathiazole) to reduce microbiome folate production since it was not mentioned. Were wire bottom cages used to eliminate coprophagy? Were there any significant differences between male and female serum folate levels that could have contributed to the endpoints. Was only a subset of samples assayed for total folate? (fig 2b

shows a possible n of 6 per group?). If no antibiotics and no wire bottom cages were used, mice can maintain adequate folate levels from coprophagy without developing signs of anemia. Please discuss these details as it helps clarify the conditions used.

RESPONSE: Excellent points, and we have now added this information (see Material and Methods):

“We note that when designing experiments to assess the consequences of folate limitation, it is common to control both folate and choline intake to ensure that the observed effects are due to the restriction of folate (Beaudin et al., 2011) because the presence of choline can mask the effects of folate deficiency. Choline can be oxidized to betaine, which provides methyl groups for converting homocysteine to methionine, independent of the folate cycle. Choline can also be incorporated into phosphatidylcholine, a major methyl ‘sink’ in the cell, through the Kennedy pathway. Lastly, we did not use any antibiotics to interfere with the microbiome nor wire bottom cages to eliminate coprophagy. Wire bottom cages were used only in the metabolic chamber experiments.”

Were there any significant differences between male and female serum folate levels that could have contributed to the endpoints. Was only a subset of samples assayed for total folate? (fig 2b shows a possible n of 6 per group?).

RESPONSE: Regarding folate levels, no significant sex differences were observed. We assayed all the animals we had at 120 weeks of age, the euthanasia endpoint, as shown in Figure 2B. There were fewer females than males in both diets.

8. There are instances in the results section where statements were made implying that there are differences observed "slightly higher", "negative association" when it is not statistically significant. There can be either statistically significant differences/correlation or not. please be precise in your wording.

RESPONSE: We have revised the Results section to ensure that qualitative descriptions such as "slightly higher" are only used when supported by appropriate statistical evidence. We have listed all the relevant numbers in each case after performing thorough and robust statistical analyses. We note, however, that mentioning qualitative descriptors is not always unwarranted, as long as they are factual.

9. Graying was observed less significantly in the F/C- group according to the authors. However, no quantitative assessment was made, and it is merely observational.

RESPONSE: It is not clear how to quantify graying non-invasively. Hence, we simply took photographs.

Inference to inhibition of mTOR was made, but mTOR protein and phosphorylation levels were not performed. The authors did perform western blotting on ribosomal S6 protein, however no assessment of the downstream mTOR targets P70S6k1 and 4EBP are shown.

RESPONSE: This is a good suggestion. We added a new experiment, looking at 4EBP1 phosphorylation (see new Figure S2). The results mirror those looking at S6 phosphorylation.

10. Can the change in RER in F/C- mice compared to controls be explained by the increased adiposity in these animals?

RESPONSE: We do not know. The relationship between adiposity and respiratory exchange rate can be quite complex. The increased adiposity of male mice limited for folate may lead to higher RER, reflecting perhaps a greater reliance on carbohydrate metabolism. But this is very speculative, especially since these mice are not obese. It is unclear how the improved metabolic plasticity could be associated with adiposity for the females.

11. How was the microbiome normalized between groups prior to the beginning of the experiment? (fecal slurry gavage, bedding exchange, cohabitation, none of the above?). There is no mention of this crucial step in the materials and methods section. Furthermore, additional details regarding the microbiome analysis are required (analysis pipeline, read depth, denoising, software, data processing, PCA analysis, etc...). it is not sufficient to state that Zymo performed the analysis.

RESPONSE: We now revised the text and added a detailed description of the methods, as follows:

“There was no microbiome normalization between groups prior to the beginning of the experiment. Mouse fecal pellets were gathered by positioning the mice on a paper towel beneath an overturned glass beaker. A minimum of three fecal pellets from each animal were transferred into cryovials using sterile forceps. The samples were preserved at -80°C and shipped to Zymo Research, where they were processed and analyzed with the ZymoBIOMICS® Shotgun Metagenomic Sequencing Service (Zymo Research, Irvine, CA). For DNA extraction, the ZymoBIOMICS®-96 MagBead DNA Kit (Zymo Research, Irvine, CA) was used according to the manufacturer’s instructions. Genomic DNA samples were profiled with shotgun metagenomic sequencing. Sequencing libraries were prepared with Illumina® DNA Library Prep Kit (Illumina, San Diego, CA) with up to 500 ng DNA input following the manufacturer’s protocol using unique dual-index 10 bp barcodes with Nextera® adapters (Illumina, San Diego, CA). All libraries were pooled in equal abundance. The final pool was quantified using qPCR and TapeStation® (Agilent Technologies, Santa Clara, CA). The final library was sequenced on the NovaSeq® (Illumina, San Diego, CA) platform. The ZymoBIOMICS® Microbial Community DNA Standard (Zymo Research, Irvine, CA) was used as a positive control for each library

preparation. Negative controls (i.e. blank extraction control, blank library preparation control) were included to assess the level of bioburden carried by the wet-lab process. Raw sequence reads were trimmed to remove low quality fractions and adapters with Trimmomatic-0.33 (Bolger et al., 2014): quality trimming by sliding window with 6 bp window size and a quality cutoff of 20, and reads with size lower than 70 bp were removed. Antimicrobial resistance and virulence factor gene identification was performed with the DIAMOND sequence aligner (Buchfink et al., 2015). Microbial composition was profiled with Centrifuge (Kim et al., 2016) using bacterial, viral, fungal, mouse, and human genome datasets. Strain-level abundance information was extracted from the Centrifuge outputs and further analyzed to perform alpha- and beta-diversity analyses and biomarker discovery with LEfSe (Segata et al., 2011) with default settings ($p > 0.05$ and LDA effect size > 2)."

What is an "easily distinguishable gut microbiome" and "appeared less diverse"?

RESPONSE: To clarify these points, we now edited as follows:

"The different sex and diet groups had an easily distinguishable gut microbiome, occupying different areas of principal component analysis graphs (Figure 5A), based on Bray-Curtis β -diversity dissimilarity indices (Knight et al., 2018). The intestinal microbiome of male mice on the F/C- diet was not statistically less diverse ($p=0.222$, based on the Wilcoxon rank sum test; Figure 5 - Supplement 1)."

a two-dimensional plot using two principal components would be more suitable for image 5A and allow for better visualization of the clustering of the groups.

RESPONSE: We tried displaying the data on a multipanel (3 panels per group, 12 total) two-dimensional figure, but the result is more confusing. Since the sample number is small ($n=6$ animals per group), the 3D graphs are visually adequate and more pleasing. They are also the standard way of representing this kind of data.

Since the authors suggest that the microbiome could be a source of 1C metabolites (including natural folate), it is important to clarify if coprophagy is involved.

RESPONSE: We agree and have added the information as requested.

12. How are inflammatory cytokines and marker levels linked to reduced anabolism and immune function in non-challenged animals?

RESPONSE: We do not make any claims for such links if that is what the reviewer implied. If the intent was more towards speculation, we suspect one could imagine various situations. For instance, nutrients may be more heavily used during inflammation to support immune cell responses instead of central anabolic processes in other tissues, limiting the building blocks available for tissue growth and repair. Since we do not see

major changes in inflammatory cytokines, we prefer not to speculate about possible links.

13. When discussing the epigenetic analysis, the authors state "no changes in the DNA methylation from liver samples.." and "groups appear younger than expected". Please clarify these statements. Additional details are needed regarding the analysis performed and the choice of methylated loci and methods. Please reference the epigenetic clock or model that was used and if was developed for the same strain and sub-strain of mice. Is it using a modified "Hovarth" mouse DNA age epigenetic clock? If so, provide the necessary details and a possible explanation for the discrepancy other than "unknown reasons"

RESPONSE: The assay is based on the "Hovarth" mouse DNA age epigenetic clock, for the strain we used (C57BL/6). We have now added a detailed description, which we received from the company, as follows (see Materials and Methods):

"Liver samples (~15mg) collected at euthanasia were placed in 0.75mL of 1X DNA/RNA Shield™ solution (Zymo Research, Irvine, CA), shipped to Zymo Research, and processed with DNAge® Service according to their established protocols. Briefly, after DNA extraction, the EZ DNA Methylation-Lightning Kit (Zymo Research, Irvine, CA) following the standard protocol was used for bisulfite conversion. Samples were enriched specifically for the sequencing of >1000 age-associated gene loci using Simplified Whole-panel Amplification Reaction Method (SWARM®), where specific CpGs are sequenced at minimum 1000X coverage. Sequencing was run on an Illumina NovaSeq instrument. Sequences were identified by Illumina base calling software then aligned to the reference genome using Bismark. Methylation levels for each cytosine were calculated by dividing the number of reads reporting a "c" by the number of reads reporting a "C" or "T". The percentage of methylation for these specific sequences were used to assess DNA age according to Zymo Research's proprietary DNAge® predictor which had been established using elastic net regression to determine the DNAge®."

As for a possible explanation for the discrepancy, since all our "groups appear younger than expected," unfortunately, other than "unknown reasons," we have none to offer. Nonetheless, the critical point for this study is that we saw no diet effects, regardless of where the company's assay draws the baseline.

14. Regarding Uracil misincorporation, the liver contains significant stores of folate as it is the main hub for several critical OCM reactions (Phospholipid methylation is a major one). Earlier studies used antibiotics with or without coprophagy prevention measures to induce a state of folate depletion to induce uracil incorporation in various tissues of rodent models. There is some controversy whether dietary folic acid restriction/methyl donor restriction alone will lead to uracil misincorporation when there is no apparent depletion or anemia. Please discuss your specific experimental procedures and how it agrees or disagrees with the published literature.

RESPONSE: We have now added the experimental details, as suggested in a previous comment. Since we do not see uracil misincorporation, we prefer not to comment on the published literature for possible links between misincorporation and anemia.

15. The section discussing RPS6 needs to be rewritten and it is difficult to understand.

RESPONSE: We revised the text, which now reads:

“Immunoblot analysis of liver tissue samples gathered at the time of euthanasia revealed variability in the detected values across individual mice. When examining the male mice, we observed that, on average, those fed the F/C- diet had approximately half the amount of phosphorylated RPS6 (P-RPS6) compared to those on the F/C+ diet. However, due to high variability in the measured values, the overall differences in P-RPS6 levels between the two dietary groups did not reach statistical significance (Figure 7 - Supplement 1; $p > 0.05$, based on the Wilcoxon rank sum test).”

Furthermore, as stated previously, considering phosphorylation of mTOR and its downstream targets 4EBP and S6K1 will give a clear indication of proliferative signaling.

RESPONSE: As we mentioned above, we have now added the suggested 4EBP experiment (see new Figure S2).

Additionally, these pathways are impacted by feeding status, diurnal cycles, and sex. Were these factors controlled prior to sacrifice? Were the animals sacrificed at the same time? In a fed or unfed state?

RESPONSE: The animals were sacrificed at the same time, with no feeding limitations.

16. The western blots provided in supplementary files show uneven protein loading across lanes (ponceau stain). No loading control is shown such as B-actin. A separate blot is used for total and phosphorylated proteins as opposed to gently stripping the membrane of the phosphorylated blot and re-incubating with the antibody for total. While normalizing phosphorylated to total protein levels will eliminate some of the variability in the author's method. The uneven loading may introduce errors in the calculated ratios.

RESPONSE: The uneven loading across mouse samples is inconsequential. We report the ratio of phospho-RPS6 to the total amount of RPS6 *within* each mouse sample. These ratios were then compared among the different animals and diet groups. We also note that stripping could introduce other artifacts if it is not uniform across all the blot areas.

17. While the authors referenced older studies utilizing low dose methotrexate on rodents and provided a composite lifespan based on these findings, why was dietary folate and choline restriction used instead of a low dose methotrexate in mice in the current study? Please provide a rationale for this approach.

RESPONSE: First, in the context of current folate fortification policies, we reasoned that testing dietary folate limitation late in life would be more informative. Second, three of us (M.P., B.K.K., and M.K.) proposed to the Interventions Testing Program at the National Institutes of Health to test whether low-dose methotrexate extends lifespan in mice. The proposal was accepted, and the study is ongoing (the ITP decided to test methotrexate at 0.2ppm, starting at 14 months of age; <https://www.nia.nih.gov/research/dab/interventions-testing-program-itp/supported-interventions>).

Minor comments:

1. While the authors make compelling arguments that lower folate intake later in life may promote healthy aging, an important consideration in the human population that a considerable percentage of older individuals may be consuming an excessive amount of folate due the combination of fortification and voluntary supplementation. An alternate hypothesis that could apply to humans and lab models is that the existing levels of exposure to folate/folic acid may be accelerating the aging process and promoting disease in later life.

RESPONSE: Perhaps, but as we describe in the text (2nd paragraph in the introduction):

“...analyses ‘did not identify specific risks from existing mandatory folic acid fortification’ in the general population (Field and Stover, 2018). This conclusion neither refutes nor contradicts the idea that a moderate decrease in folic acid intake among older adults may improve healthspan. Merely because high folic acid intake does not harm the health of older adults does not negate the possibility that a lower folic acid intake might enhance health.”

2. The common C57BL/6j is being referred to as the "long lived strain". Is this relative to mice in wild conditions? There are many transgenic C57bl/6 strains that live considerably longer. Please clarify if this is meant to describe the aged mice used in the experimental process.

RESPONSE: This was from a comprehensive comparison of many different inbred strains. We apologize for omitting the citation, which we have now added (Yuan et al, 2009).

3. While the authors state early in the manuscript that longevity was not a measured outcome in the mouse study, the manuscript contains statements discussing animal survival in the results and survival curves (figure 2). This gives the impression that the study was planned as a survival analysis initially and since no difference was observed between the experimental groups during the earlier stages, the secondary endpoints of health span analysis were adopted. Either approach does not detract from the significance of the study's findings. Further clarity on the approach would be beneficial to the readers.

RESPONSE: The study was designed, and the Animal Use Protocol was institutionally approved for healthspan, not lifespan. The number of animals we used did not have

sufficient power to detect lifespan differences. Note that, at least for males, very few animals had died by 120 weeks, our approved euthanasia endpoint. However, it was important to report that folate limitation did not adversely affect overall survival during the analysis time frame.

4. For yeast culture conditions, what are the folate sources and content? Is there added folic acid similar to cell culture conditions where supraphysiological concentrations are used in standard mediums (RPMI and DMEM).

RESPONSE: The yeast media we used were undefined (YPD, see Materials and Methods). The source of folate in this media is “yeast extract,” which is generally considered to contain very high amounts of folate (it was used decades ago to treat anemia and folate deficiency in pregnant women). Note also that, unlike animals, yeast can synthesize folate.

5. In the metabolism section, the authors make statements such as "the differences were minimal" , "probably were due..", "minimal effects", "apparent increase", "tended to be", "little uracil" etc.. please refrain from using subjective language and use precise scientific terms.

RESPONSE: Please see our earlier response to this comment.

6. Figure 2-c, there is a typo, Weeks not months

RESPONSE: Corrected. Thank you!

** Referees cross-commenting**

while we generally agree with the other reviewer's concerns, we find that reviewer 3 rejection of the authors conclusion without considering the evidence presented in the context of what is currently known in the field potentially limiting. Multiple groups have shown that manipulation of OCM enzymes (DHFR, TYMS, SAMS) can extend lifespan in worms. the recent report Antebi's group (Annibal et al. Nature Com, 2021) provides strong evidence that OCM is central to longevity regulation in worms and mice and that folate intake can interact with and modulate organismal longevity. while this manuscript findings are not conclusive, I think it is premature to dismiss it completely. perhaps the alternative is to discuss the limitations of this approach and interpret the results (or the lack of significant differences) in order to help guide future research into this important subject. generalizing rodent results to human is always going to be a limiting factor in this type of work. Mice have significantly higher circulating folate. additionally, DHFR activity (the rate limiting enzyme in folate OCM) in rodents can be up to 100 times higher than its human equivalent. another consideration is that mice, similar to other rodents, engage in coprophagy, thereby recycling and supplementing bacterially produced folate in the absence of antibiotics in the diet. Therefore, mice placed of dietary folate restriction in the absence of antibiotics do not develop signs of anemia or deficiency. Therefore, it could be argued that there

is no loss of nutrients in mice in this scenario and that supplementation at the arbitrarily recommended level of synthetic folic acid (2mg/kg day) or higher could impact health and aging. Similarly, in humans excess folate intake has been controversially associated with a number of deleterious health effects. It is important not to dismiss these reports and encourage further research into this subject that impacts a significant percentage of the human population due to the widespread use of supplements.

RESPONSE: We thank the reviewers for their evaluation of the work we presented. We have also added the following in the discussion, expanding the limitations of the study:

“Since mice engage in coprophagy, microbiome contributions to folate metabolism are bound to be substantial in this species. There are also significant differences in folate status between mice and people. For example, people have lower levels (~10-15 ng/mL) of serum folate than mice (Bailey et al., 2015), and the activity of DHFR, an enzyme essential for maintaining tetrahydrofolate pools -the folate form used in 1C reactions, maybe only 2% of that in rodents (Bailey and Ayling, 2009). Hence, mice are likely more refractory to a low folate dietary intake.”

Reviewer #1 (Significance (Required)):

Significance:

A major strength of this study is that the authors show that manipulation of OCM either through pharmacological inhibition or dietary restriction can impact organismal longevity in a conserved manner across species from yeast to worms and mammals. These findings provide compelling evidence that folate intake and metabolism in humans should be rigorously researched as potential regulator of aging. These findings complement and agree with a recent report by Antebi's group (Annibal et al. Nature Com, 2021) highlighting that long-lived worm and mice strains exhibit similar metabolic regulation of one carbon metabolism. In the same report low levels of folate supplementation partially or completely abrogated the lifespan extension in some models. This study provides additional evidence that restricting OCM through drugs or dietary restriction can significantly impact healthspan and lifespan. Additionally, it raises the question whether excessive folate intake in aged adults may have potentially deleterious effects on health and longevity. The limitations of this study can be seen in the overall lack of significant impact of the dietary intervention on the health metrics that were measured in mice. The study does not provide strong evidence that restricting folate and choline intake will produce favorable effects on health. Similarly, no significant impact on mice lifespan was observed based on the partial lifespan analysis. Further clarity is needed regarding the experimental procedures and methods used. The study, nonetheless, is an important step towards investigating the role of folate and OCM in regulating mammalian healthspan and lifespan. Future studies can expand on these findings and investigate whether OCM interventions that are started in early life can produce significant and measurable effects on longevity and health in mammals. The findings here provide a conceptual and incremental advance in our understanding of these complex interactions.

These findings are important to the research communities especially in the areas of longevity, metabolism, and nutrition.

RESPONSE: We appreciate the recognition of our work's significance in furthering understanding of longevity, metabolism, and nutrition. We would also like to stress that this study is not an incremental advance. We believe our study's focus on dietary folate limitation in aged mice represents a novel and more radical contribution, considering the lack of prior research in this specific context, underscoring the distinctiveness and importance of our findings.

Reviewer #2 (Evidence, reproducibility and clarity (Required)):

Summary: In this manuscript they investigate whether disruption of the folate cycle can slow ageing/improve health in yeast, worms and mice. There are a few experiments in yeast and *C. elegans* but the rest is a meta analysis of some old data on folate-deprived mice and their own study of mice on a diet with and without folic acid and choline. They find that various interventions of the folate cycle extend lifespan in yeast and worms, that the old study suggest mice live longer without folic acid supplementation and that there is no change to healthspan with mice without folic acid and choline in the diet late in life and that these mice show some positive benefits. Analysis of the microbiome and the transcriptomics suggest small changes to the microbiota and changes in gene expression. Overall the authors conclude that biosynthetic processes have been inhibited without negative effects on healthspan.

Major comments

1. The two worm lifespan experiments in Fig 1 show very different controls despite the methods stating that the conditions were the same. Controls can vary from one experiment to another but the difference is striking. It would be good to have supplementary data about the number of repeats and other data about these experiments.

RESPONSE: We also noted the difference. However, we believe our conclusions are valid and robust because we used only experiment-matched controls for each comparison. We now describe in detail how the experiments were done (see revised Materials and Methods). Lastly, the two compounds were tested years apart from different individuals, and the different lifespans of the controls could arise from differences in the media batches, temperature control, etc.

2. The diet lack folic acid and choline yet the conclusions are only about folate. The choline aspect of the diet needs to be acknowledged as a potential factor.

RESPONSE: As we mentioned above, we have now added this information (see Material and Methods):

“We note that when designing experiments to assess the consequences of folate limitation, it is common to control both folate and choline intake to ensure that the

observed effects are due to the restriction of folate (Beaudin et al., 2011) because the presence of choline can mask the effects of folate deficiency. Choline can be oxidized to betaine, which provides methyl groups for converting homocysteine to methionine, independent of the folate cycle. Choline can also be incorporated into phosphatidylcholine, a major methyl 'sink' in the cell, through the Kennedy pathway. Lastly, we did not use any antibiotics to interfere with the microbiome nor wire bottom cages to eliminate coprophagy. Wire bottom cages were used only in the metabolic chamber experiments.”

3. The authors argue that the effects on the mice are not mediated effects on the diet by the microbiome because there is not a statistical effect on diversity. However they do show a clear difference at the metagenomic level that fits with a metabolic difference. It also ignores work in *C. elegans* showing that inhibition of bacterial folate synthesis increases lifespan, not by decreasing folate supply but because lowered bacterial folate prevents an age-accelerating activity in the bacteria (Virk et al 2016). It has also been shown that a breakdown product of folic acid can be taken up by bacteria and influence ageing (Maynard et al 2018). I do not think the evidence is strong enough to discounted that the changes seen in the mice are not mediated by microbes.

RESPONSE: We do not state that “changes seen in the mice are not mediated by microbes”. On the contrary, we agree with the reviewer that the microbiome likely contributes significantly, and we hope this is conveyed in the text. We also agree with the references the reviewer pointed out, which we cite (see also our response to point#5 of reviewer 1).

Minor comments

1. It had been shown a long time ago that *sams-1* mutants in *C. elegans* extend lifespan. MTX is likely to influence SAMS levels. This point needs to be mentioned.

RESPONSE: Thank you. We added the reference.

2. Page - 6 "folate accelerates worm aging". This statement is not correct and is not what Virk et al 2016 suggests.

RESPONSE: We revised it to the following: “It has been reported that treating worms with high levels of methotrexate (220µM) at the adult stage did not extend their lifespan (Virk et al., 2016)”.

3. Page 7. "at 100µM, a dose similar to the one used in mice with metabolic syndrome (Asby et al., 2015)." It's not valid to compare the concentration of a drug in the media in a *C. elegans* experiment to a dose given to mice.

RESPONSE: We appreciate the reviewer's point on comparing drug dosages across species. The intention was to provide a reference point for the concentration used rather than suggesting a direct equivalence with outcomes. We recognize the complexities of cross-species dosage comparisons and have amended the text to clarify that the mention of dosage is for contextual purposes only.

** Referees cross-commenting**

I would like to add that it is important to consider whether there are in fact negative effects of folic acid given in later life and this is one of the only studies that addresses this question in a mammalian model, and thus needs to be reported, once the issues raised have been addressed.

RESPONSE: As we mentioned in a comment from reviewer 1 and describe in the text (2nd paragraph in the introduction):

“...analyses ‘did not identify specific risks from existing mandatory folic acid fortification’ in the general population (Field and Stover, 2018). This conclusion neither refutes nor contradicts the idea that a moderate decrease in folic acid intake among older adults may improve healthspan. Merely because high folic acid intake does not harm the health of older adults does not negate the possibility that a lower folic acid intake might enhance health.”

Reviewer #2 (Significance (Required)):

The main strength of this manuscript is that it examines the effect of mice given a folate and choline deficient diet late in life and finds mostly positive effects. This finding challenges the dogma that folate

Reviewer #3 (Evidence, reproducibility and clarity (Required)):

Blank/Polymenis and colleagues explore how reduced folate metabolism impacts aging. While folate supplementation is known to benefit the development and health of young people, little is known about the impact of this substrate at advanced ages. The paper consists of two parts: 1) blocking folate metabolism in yeast and *C. elegans* while measuring lifespan (reproductive or age of death); 2) measuring a vast array of traits in mice where folate (and choline) is removed from the diet starting at age 1 year. The second approach is most central to the paper's theme, and the authors conclude their 'data raise the exciting possibility that ... reduced folate intake later in life might be beneficial.' However, I do not accept this conclusion. Instead, the overwhelming fact is that there were no changes in any phenotype due to the absence of F/C in the older animals. Loss of this nutrient is neutral, although perhaps bad for the kidney. In my view, the

authors misinterpret their very basic results: loss of dietary folate has no impact on aged mice (one strain, at that). And there is no way to generalize this simple conclusion to humans.

RESPONSE: We respectfully disagree with the reviewer's assessment of our study's conclusions and its significance. With the primary focus on evaluating the effects of reduced folate intake in aged mice, we explored a comprehensive range of healthspan markers and molecular analyses. Contrary to the reviewer's assertion, our data demonstrate significant outcomes such as altered body weight and metabolic parameters in mice subjected to folate restriction, along with insights into molecular changes indicative of lower anabolism.

The reviewer's interpretation that folate limitation has no observable impact on aged mice overlooks the nuanced findings presented in our study. While acknowledging the neutral effects observed in some phenotypes, we contend that our results collectively contribute to a deeper understanding of the implications of late-life folate restriction. It is unwarranted to dismiss these findings.

Generalizing findings from model systems to humans is indeed complex, as noted by the reviewer. However, our study, alongside existing literature, provides valuable insights that warrant consideration and further exploration. We stand by the rigor of our methodology, the diversity of data presented, and the significance of our results in enhancing knowledge on the impact of folate metabolism in aging models.

There are other issues throughout the work that need to be addressed but given weakness on its key argument, I will not elaborate these points.

RESPONSE: Since the reviewer offered no specifics on "other issues," we cannot respond. We hope, however, that we have addressed them in our response to the other reviewers' comments.

Reviewer #3 (Significance (Required)):

Blank/Polymenis and colleagues explore how reduced folate metabolism impacts aging. While folate supplementation is known to benefit the development and health of young people, little is known about the impact of this substrate at advanced ages.

RESPONSE: We concur with the reviewer's observation regarding the knowledge gap surrounding the impact of reduced folate metabolism on aging, particularly in advanced stages of life, which is why our study significantly contributes to the field. As we mentioned above, not only do we report that some healthspan metrics were improved in folate-limited animals (e.g., body weight, improved metabolic plasticity), but our study also offers for the first time a comprehensive biomarker analysis of folate limitation late in life (e.g., metabolite and mRNAs changes associated with lower anabolism, lower IGF1

levels in females). This original contribution enhances our understanding of the complex interplay between folate metabolism and aging.

June 28, 2024

RE: Life Science Alliance Manuscript #LSA-2024-02868

Dr. Michael Polymenis
Texas A&M University
Biochemistry and Biophysics
2128 TAMU
College Station, Texas 77843

Dear Dr. Polymenis,

Thank you for submitting your revised manuscript entitled "Late-life dietary folate restriction reduces biosynthetic processes without compromising healthspan in mice". We would be happy to publish your paper in Life Science Alliance pending final revisions necessary to meet our formatting guidelines.

- please be sure that the authorship listing and order is correct
- please upload all figure files as individual ones, including the supplementary figure files; all figure legends should only appear in the main manuscript file
- please upload your main manuscript text as an editable doc file
- please add a Running Title and a Summary Blurb/Alternate Abstract to our system
- please add a Category for your manuscript in our system
- please add the Twitter handle of your host institute/organization as well as your own or/and one of the authors in our system
- please add an Author Contributions section to your main manuscript text, and in our system
- please add a Conflict of Interest statement to your main manuscript text
- please use the [10 author names et al.] format in your references (i.e., limit the author names to the first 10)
- please add your main and supplementary figure legends to the main manuscript text after the references section and remove them from figures
- Label supplementary figures as Figures S1, S2, etc., in the callouts in the manuscript text. There cannot be more supplements under one figure. For example, Figure S2 cannot have subimages. Instead, upload supplementary figures like you did for the main figures. Figure S2 can have panels A and B, but supplement 2, for Figure 2, should be figure S3 with panels A and B, etc
- please add a callout for Figure 2C to your main manuscript text
- please be sure to cite all panels of the supplementary figures
- please add a Data Availability statement at the end of the Materials and Methods that include the accession information for the RNA-seq data

A. FINAL FILES:

B. MANUSCRIPT ORGANIZATION AND FORMATTING:

Sincerely,

Reviewer #1 (Comments to the Authors (Required)):

the manuscript has been improved significantly and the authors adequately addressed my previous concerns. no further revisions are necessary. The work presented by the authors is an important contribution to the field and will help improve our understanding of the role folate/one carbon metabolism in mammalian aging and health.

Reviewer #2 (Comments to the Authors (Required)):

I am happy that the issues I raised have been addressed satisfactorily by the authors and I support publication of this manuscript.

July 1, 2024

RE: Life Science Alliance Manuscript #LSA-2024-02868R

Dr. Michael Polymenis
Texas A&M University
Biochemistry and Biophysics
2128 TAMU
College Station, Texas 77843

Dear Dr. Polymenis,

Thank you for submitting your Research Article entitled "Late-life dietary folate restriction reduces biosynthesis without compromising healthspan in mice". It is a pleasure to let you know that your manuscript is now accepted for publication in Life Science Alliance. Congratulations on this interesting work.

DISTRIBUTION OF MATERIALS:

Again, congratulations on a very nice paper. I hope you found the review process to be constructive and are pleased with how the manuscript was handled editorially. We look forward to future exciting submissions from your lab.

Sincerely,
